# VARIANCE REDUCED LOCAL SGD
# WITH LOWER COMMUNICATION COMPLEXITY

## ABSTRACT

To accelerate the training of machine learning models, distributed stochastic gradient descent (SGD) and its variants have been widely adopted, which apply multiple workers in parallel to speed up training. Among them, Local SGD has gained much attention due to its lower communication cost. Nevertheless, when the data distribution on workers is non-identical, Local SGD requires $O(T^{\frac{3}{4}}N^{\frac{3}{4}})$ communications to maintain its *linear iteration speedup* property, where $T$ is the total number of iterations and $N$ is the number of workers. In this paper, we propose Variance Reduced Local SGD (VRL-SGD) to further reduce the communication complexity. Benefiting from eliminating the dependency on the gradient variance among workers, we theoretically prove that VRL-SGD achieves a *linear iteration speedup* with a lower communication complexity $O(T^{\frac{1}{2}}N^{\frac{3}{2}})$ even if workers access non-identical datasets. We conduct experiments on three machine learning tasks, and the experimental results demonstrate that VRL-SGD performs impressively better than Local SGD when the data among workers are quite diverse.

## 1 INTRODUCTION

With the expansion of data and model scale, the training of machine learning models, especially deep learning models has become increasingly time-consuming. To accelerate the training process, distributed parallel optimization has attracted widespread interests recently, which encourages multiple workers to cooperatively optimize the model.

For large-scale machine learning problems, stochastic gradient descent (SGD) is a fundamental tool. It can be easily parallelized by collecting stochastic gradient from different workers and hence it is widely adopted. Previous studies (Dekel et al., 2012; Ghadimi & Lan, 2013) justify that synchronous stochastic gradient descent (S-SGD) has a *linear iteration speedup* for both general convex and non-convex objectives, which means that the total number of iterations is reduced by $N$ times with $N$ workers. However, S-SGD suffers from a major drawback: the communication cost among workers is expensive when the number of workers is large, which prevents S-SGD from achieving a *linear time speedup*. Therefore, it is crucial to overcome the communication bottleneck.

To reduce communication cost, several studies (Wang & Joshi, 2018; Zhou & Cong, 2018; Stich, 2019; Yu et al., 2019b; Shen et al., 2019) have managed to lower the communication frequency. Among them, Local SGD (Stich, 2019) is a representative distributed algorithm, where workers can conduct SGD locally and average model with each other every $k$ iterations. Compared with S-SGD, the algorithms based on Local SGD reduce the communication rounds from $O(T)$ to $O(T/k)$. To deal with the gradient variance among workers, previous studies require at least one of the following extra assumptions: (1) the bounded gradient variance among workers; (2) an upper bound for gradients; (3) identical data on all workers. When the data distribution on workers is identical, which is the so-called *identical case*, the algorithms based on Local SGD can exhibit superior performance. Nevertheless, the identical data assumption is not always valid in real cases. When the data distribution on workers is non-identical, which is the so-called *non-identical case*, these algorithms would encounter a significant degradation in the convergence rate due to the gradient variance among workers. We seek to eliminate the gradient variance among workers, which may make the algorithm converge much faster than the vanilla Local SGD.

In this paper, we propose Variance Reduced Local SGD (VRL-SGD), a novel distributed optimization algorithm to further reduce the communication complexity. Benefiting from an additional vari-

ance reduction component, VRL-SGD eliminates the extra assumption about bounded gradient variance among workers in previous studies based on Local SGD (Yu et al., 2019a;b; Shen et al., 2019). Thus the communication complexity can be reduced from $O(T^{\frac{3}{4}}N^{\frac{3}{4}})$ to $O(T^{\frac{1}{2}}N^{\frac{3}{2}})$ in VRL-SGD for the *non-identical case*, which is crucial for achieving a better time speedup. Therefore, VRL-SGD is more suitable than Local SGD for the scenarios, such as federated learning (Konečný et al., 2016), where the gradient variance across workers might be large.

Contributions of this paper are summarized as follows:

- We propose VRL-SGD, a novel distributed optimization algorithm with a state-of-the-art communication complexity. Specifically, the communication complexity is reduced from $O(T^{\frac{3}{4}}N^{\frac{3}{4}})$ to $O(T^{\frac{1}{2}}N^{\frac{3}{2}})$ for the *non-identical case*. To the best of our knowledge, this is the first time that an algorithm based on Local SGD possesses such a communication complexity for the *non-identical case*. Meanwhile, VRL-SGD also achieves the optimal communication complexity for the *identical case*.

- We provide a theoretical analysis and prove that VRL-SGD has a *linear iteration speedup* with respect to the number of workers. Our method does not require the extra assumptions, e.g. the gradient variance across workers is bounded.

- We validate the effectiveness of VRL-SGD on three standard machine learning tasks. And experimental results show that the proposed algorithm performs significantly better than Local SGD if data distribution in workers is different, while maintains the same convergence rate as Local SGD if all workers access identical datasets.

## 2 RELATED WORK

Synchronous stochastic gradient descent (S-SGD) is a parallelized version of mini-batch SGD and is theoretically proved to achieve a *linear iteration speedup* with respect to the number of workers (Dekel et al., 2012; Ghadimi & Lan, 2013). Nevertheless, due to the communication bottleneck, it is difficult to obtain the property of *linear time speedup*. To eliminate communication bottlenecks, many distributed SGD-based methods are proposed, such as lossy compression methods (Alistarh et al., 2017; Aji & Heafield, 2017; Bernstein et al., 2019; Lin et al., 2018b; Karimireddy et al., 2019; Tang et al., 2019), which use inexact approximations or partial data to represent the gradients, and methods (Stich, 2019; Yu et al., 2019b) based on the lower communication frequency.

Among them, Local SGD (Stich, 2019), a representative method to lower the communication frequency, has been widely used in the training of large-scale machine learning models, and its superior performance is verified in several tasks (Povey et al., 2014; Su & Chen, 2015; Lin et al., 2018a). In Local SGD, each worker conducts SGD updates locally and averages its model with others periodically. Previous studies have proven that Local SGD can attain a *linear iteration speedup* for both strongly convex (Stich, 2019) and non-convex (Yu et al., 2019b) problems. To fully utilize hardware resources, a variant of Local SGD, called CoCoD-SGD (Shen et al., 2019), is proposed with the decoupling of computation and communication. Furthermore, Yu et al. (2019a) provide a clear linear speedup analysis for Local SGD with momentum. However, most of the above algorithms assume that the gradient variance among workers is bounded, and some of them even depend on a stronger assumption, e.g., the data distribution on workers is identical. Dependence on these assumptions may lead to a slow convergence rate for the *non-identical case*, which limits the further reduction of communication frequency and avoids a better time speedup. Haddadpour et al. (2019) verify that the use of redundant data can lead to lower communication complexity and hence faster convergence. The redundant data can help reduce the gradient variance among workers, thus it avoids the slow convergence rate. Nevertheless, this method may be constrained in some cases. For instance, it could not be widely applied in federated learning (Konečný et al., 2016) as data cannot be exchanged between workers for privacy-preserving.

Although there are many studies proposed to reduce the variance in SGD, e.g., SVRG (Johnson & Zhang, 2013), SAGA (Defazio et al., 2014), and SARAH (Nguyen et al., 2017), they could not directly deal with the gradient variance among workers in distributed optimization. In recent years, several studies (Shi et al., 2015; Mokhtari & Ribeiro, 2016; Tang et al., 2018) have proposed to eliminate the gradient variance among workers in the decentralized setting. Among them, Shi et al. (2015) propose a novel decentralized algorithm, EXTRA, which provides an ergodic convergence

Table 1: Comparisons of the communication complexity for different algorithms. The second column and the third column show communication complexity for identical and non-identical datasets respectively. Here, we regard the following assumptions as extra assumptions: (1) an upper bound for gradients; (2) the bounded gradient variance among workers.

| REFERENCE | IDENTICAL DATA | NON-IDENTICAL DATA | EXTRA ASSUMPTIONS |
|---|---|---|---|
| GHADIMI & LAN (2013) | $T$ | $T$ | NO |
| YU ET AL. (2019B) | $O(N^{\frac{3}{4}}T^{\frac{3}{4}})$ | $O(N^{\frac{3}{4}}T^{\frac{3}{4}})$ | (1) |
| SHEN ET AL. (2019) | $O(N^{\frac{3}{2}}T^{\frac{1}{2}})$ | $O(N^{\frac{3}{4}}T^{\frac{3}{4}})$ | (2) |
| THIS PAPER | $O(N^{\frac{3}{2}}T^{\frac{1}{2}})$ | $O(N^{\frac{3}{2}}T^{\frac{1}{2}})$ | NO |

rate for convex problems and a linear convergence rate for strongly convex problems benefiting from eliminating the variance among workers. The $D^2$ (Tang et al., 2018) algorithm further applies the variance reduction on non-convex stochastic decentralized optimization problems and removes the impact of the gradient variance among workers on the convergence rate.

To eliminate the gradient variance among workers and accelerate the training, we incorporate the variance reduction technique into Local SGD, and hence reduce the extra assumptions in the theoretical analysis. For a better comparison with related algorithms in terms of communication complexity and assumptions, we summarize the results in Table 1. It presents that our algorithm achieves better communication complexity compared with the previous algorithms for the *non-identical case* and does not need extra assumptions.

## 3 PRELIMINARY

### 3.1 PROBLEM DEFINITION

We focus on data-parallel distributed training, where $N$ workers collaboratively train a machine learning model, and each worker may have its data with different distributions, which is the *non-identical case*. We use $\mathcal{D}_i$ to denote the local data distribution in the $i$-th worker. Specifically, we consider the following finite-sum optimization:

$$\min_{\boldsymbol{x} \in \mathbb{R}^d} f(\boldsymbol{x}) := \frac{1}{N} \sum_{i=1}^{N} f_i(\boldsymbol{x}), \tag{1}$$

where $f_i(\boldsymbol{x}) := \mathbb{E}_{\xi_i \sim \mathcal{D}_i}[f_i(\boldsymbol{x}, \xi_i)]$ is the local loss function of the $i$-th worker.

### 3.2 NOTATIONS

First of all, we summarize the key notations of this paper as follows.

- $\|\cdot\|$ denotes the $\ell_2$ norm of a vector.
- $f^*$ is the optimal value of equation (58).
- $\mathbb{E}$ denotes that the expectation is taken with respect to all random indexes sampled to calculate stochastic gradients in all iterations.
- $\boldsymbol{x}_i^t$ denotes the local model of the $i$-th worker at the $t$-th iteration.
- $\hat{\boldsymbol{x}}^t$ denotes the average of local models over all $N$ workers, and that is $\hat{\boldsymbol{x}}^t = \frac{1}{N} \sum_{i=1}^{N} \boldsymbol{x}_i^t$.
- $\nabla f_i(\boldsymbol{x}_i^t, \xi_i^t)$ is a stochastic gradient of the $i$-th worker at the $t$-th iteration.
- $t'$ represents the iteration of the last communication, and that is $t' = \lfloor \frac{t}{k} \rfloor k$.
- $t''$ represents the iteration of the penultimate communication, and that is $t'' = (\lfloor \frac{t}{k} \rfloor - 1)k$.

## 3.3 Assumptions

Throughout this paper, we make the following assumptions, which are commonly used in the theoretical analysis of distributed algorithms (Stich, 2019; Yu et al., 2019a; Shen et al., 2019).

**Assumption 1**

(1) **Lipschitz gradient**: *All local functions $f_i$'s have L-Lipschitz gradients*

$$\|\nabla f_i(\boldsymbol{x}) - \nabla f_i(\boldsymbol{y})\| \leq L\|\boldsymbol{x} - \boldsymbol{y}\|, \forall i, \forall \boldsymbol{x}, \boldsymbol{y} \in \mathbb{R}^d. \tag{2}$$

(2) **Bounded variance within each worker**: *There exists a constant $\sigma$ such that*

$$\mathbb{E}_{\xi \sim \mathcal{D}_i}\|\nabla f_i(\boldsymbol{x}, \xi) - \nabla f_i(\boldsymbol{x})\|^2 \leq \sigma^2, \ \ \forall \boldsymbol{x} \in \mathbb{R}^d, \forall i. \tag{3}$$

(3) **Dependence of random variables**: *$\xi_i^t$'s are independent random variables, where $t \in \{0, 1, \cdots, T-1\}$ and $i \in \{1, 2, \cdots, N\}$.*

Previous studies based on Local SGD assume that the gradient variance among workers is bounded, or even depend on a stronger assumption, e.g., an upper bound for gradients or identical data distribution on workers, while we do not require these assumptions.

## 4 Algorithm

In this section, we first introduce the proposed algorithm and then give an intuitive explanation.

### 4.1 Variance Reduced Local SGD

We propose VRL-SGD, a variant of Local SGD. VRL-SGD allows locally updating in each worker to reduce the communication cost. But there are a few more steps in VRL-SGD to eliminate the gradient variance among workers. And in VRL-SGD, a worker:

1. Communicates with other workers to get the average of all local models $\hat{\boldsymbol{x}}^t = \frac{1}{N}\sum_{i=1}^N \boldsymbol{x}_i^t$.

2. Calculates $\Delta_i^{t'}$, which denotes the average deviation of gradient between the local gradients and the global gradients in the previous period. And it is defined as

$$\Delta_i^{t'} = \Delta_i^{t''} + \frac{1}{k\gamma}(\hat{\boldsymbol{x}}^t - \boldsymbol{x}_i^t), \tag{4}$$

where $k$ is the communication period and $\gamma$ is the learning rate.

3. Updates local model $k$ times with a stochastic approximation gradient $\boldsymbol{v}_i^t$ in the form of

$$\boldsymbol{x}_i^{t+1} = \boldsymbol{x}_i^t - \gamma \boldsymbol{v}_i^t. \tag{5}$$

The essential part of equation (5) is the gradient approximation $\boldsymbol{v}_i^t$, which is formed by

$$\boldsymbol{v}_i^t = \nabla f_i(\boldsymbol{x}_i^t, \xi_i^t) - \Delta_i^{t'}. \tag{6}$$

The complete procedure of VRL-SGD is summarized in Algorithm 1. VRL-SGD allows each worker to maintain its local model $\boldsymbol{x}_i^t$ and gets the average of all local models every $k$ steps. Note that VRL-SGD with $k = 1$ is equivalent to S-SGD. While VRL-SGD with $k > 1$ reduces the number of communication rounds by $k$ times compared with S-SGD. And VRL-SGD is equivalent to Local SGD if we set $\Delta_i$ be 0 in line 5 of Algorithm 1 all the time.

To achieve a *linear iteration speedup*, Local SGD requires that $T$ is more than $O(N^3k^4)$. In other words, the communication period $k$ in Local SGD is bounded by $O(T^{\frac{1}{4}}/N^{\frac{3}{4}})$, which reduces the communication complexity to $O(N^{\frac{3}{4}}T^{\frac{3}{4}})$. Notice that a better communication period bound $O(T^{\frac{1}{2}}/N^{\frac{3}{2}})$ can be attained in the *identical case* in the previous studies (Shen et al., 2019; Yu et al., 2019a). Nevertheless, the proposed algorithm can attain the communication period bound $O(T^{\frac{1}{2}}/N^{\frac{3}{2}})$ in both the *identical case* and the *non-identical case*.

---

**Algorithm 1** Variance Reduced Local SGD (VRL-SGD)

---

1: **Input:** Initialize $\boldsymbol{x}_i^0 = \hat{\boldsymbol{x}}^0 \in \mathbb{R}^d, \Delta_i^0 = \mathbf{0} \in \mathbb{R}^d, \forall i$ and $t = 0$. Set learning rate $\gamma > 0$ and communication period $k > 0$.

2: **while** $t < T$ **do**

3:     **Worker** $W_i$ **does**:

4:     Communicate with other workers to get the average of all local models: $\hat{\boldsymbol{x}}^t = \frac{1}{N}\sum_{i=1}^N \boldsymbol{x}_i^t$.

5:     $\Delta_i^{t'} = \Delta_i^{t''} + \frac{1}{k\gamma}(\hat{\boldsymbol{x}}^t - \boldsymbol{x}_i^t)$.

6:     Update local model $\boldsymbol{x}_i^t = \hat{\boldsymbol{x}}^t$.

7:     **for** $\tau = t$ to $t + k - 1$ **do**

8:         Calculate a stochastic gradient $\nabla f_i(\boldsymbol{x}_i^\tau, \xi_i^\tau)$.

9:         $\boldsymbol{v}_i^\tau = \nabla f_i(\boldsymbol{x}_i^\tau, \xi_i^\tau) - \Delta_i^{t'}$.

10:        Each worker updates its local model:

$$\boldsymbol{x}_i^{\tau+1} = \boldsymbol{x}_i^\tau - \gamma \boldsymbol{v}_i^\tau.$$

11:     **end for**

12:     $t = t + k$.

13: **end while**

---

One might wonder why VRL-SGD can improve the convergence rate of Local SGD. VRL-SGD uses an inexact variance reduction technique to reduce the variance among workers. To better understand the intuition of VRL-SGD, let us see the update of $\Delta_i$ in equation (4). By summing up all $\Delta_i$ from 0 to $t'$ and using the fact that $\Delta_i^0 = 0$, we have

$$\Delta_i^{t'} = \frac{1}{k\gamma}\sum_{s=0}^{\lfloor\frac{t}{k}\rfloor}\left(\hat{\boldsymbol{x}}^{ks} - \boldsymbol{x}_i^{ks}\right). \tag{7}$$

By summing up the above equality over $i = 1, \cdots, N$, we obtain

$$\sum_{i=1}^N \Delta_i^{t'} = \frac{1}{k\gamma}\sum_{i=1}^N\sum_{s=0}^{\lfloor\frac{t}{k}\rfloor}\left(\hat{\boldsymbol{x}}^{ks} - \boldsymbol{x}_i^{ks}\right) = \frac{1}{k\gamma}\left(N\sum_{s=0}^{\lfloor\frac{t}{k}\rfloor}\hat{\boldsymbol{x}}^{ks} - \sum_{i=1}^N\sum_{s=0}^{\lfloor\frac{t}{k}\rfloor}\boldsymbol{x}_i^{ks}\right) = 0.$$

It shows that the expectation of $\Delta_i^{t'}$ over $i$ is zero, thus we can obtain the new update form with respect to $\hat{\boldsymbol{x}}^t$.

$$\hat{\boldsymbol{x}}^t = \hat{\boldsymbol{x}}^{t-1} - \gamma\frac{1}{N}\sum_{i=1}^N \boldsymbol{v}_i^{t-1} = \hat{\boldsymbol{x}}^{t-1} - \gamma\frac{1}{N}\sum_{i=1}^N\left(\nabla f_i(\boldsymbol{x}_i^t, \xi_i^t) - \Delta_i^{t'}\right) = \hat{\boldsymbol{x}}^{t-1} - \gamma\frac{1}{N}\sum_{i=1}^N \nabla f_i(\boldsymbol{x}_i^t, \xi_i^t). \tag{8}$$

It can be noticed that the update of $\hat{\boldsymbol{x}}^t$ in equation (8) is in the form of the generalized stochastic gradient descent. In addition, we can obtain a new representation of $\Delta_i^{t'}$ as below:

$$\begin{aligned}
\Delta_i^{t'} &= \Delta_i^{t''} + \frac{1}{k\gamma}\left(\hat{\boldsymbol{x}}^{t''} - \gamma\sum_{\tau=t''}^{t'-1}\frac{1}{N}\sum_{j=1}^N \boldsymbol{v}_j^\tau - \hat{\boldsymbol{x}}^{t''} + \gamma\sum_{\tau=t''}^{t'-1}\boldsymbol{v}_i^\tau\right) \\
&= \Delta_i^{t''} + \frac{1}{k\gamma}\left(\gamma\sum_{\tau=t''}^{t'-1}\left(\nabla f_i(\boldsymbol{x}_i^\tau, \xi_i^\tau) - \Delta_i^{t''}\right) - \gamma\sum_{\tau=t''}^{t'-1}\frac{1}{N}\sum_{j=1}^N\left(\nabla f_j(\boldsymbol{x}_j^\tau, \xi_j^\tau) - \Delta_j^{t''}\right)\right) \\
&= \frac{1}{k}\sum_{\tau=t''}^{t'-1}\left(\nabla f_i(\boldsymbol{x}_i^\tau, \xi_i^\tau) - \frac{1}{N}\sum_{j=1}^N \nabla f_j(\boldsymbol{x}_j^\tau, \xi_j^\tau)\right). \tag{9}
\end{aligned}$$

Substituting equation (9) into equation (6), we have

$$\boldsymbol{v}_i^t = \nabla f_i(\boldsymbol{x}_i^t, \xi_i^t) - \frac{1}{k}\sum_{\tau=t''}^{t'-1}\nabla f_i(\boldsymbol{x}_i^\tau, \xi_i^\tau) + \frac{1}{Nk}\sum_{\tau=t''}^{t'-1}\sum_{j=1}^N \nabla f_j(\boldsymbol{x}_j^\tau, \xi_j^\tau). \tag{10}$$

The representation of $\boldsymbol{v}_i^t$ in equation (10) can be regarded as the form of the generalized variance reduction, which is similar to SVRG (Johnson & Zhang, 2013) and SAGA (Defazio et al., 2014).

To observe that the variance among workers is reduced, we assume that the gradient variance within each worker is zero, which means that we calculate $\nabla f_i(\boldsymbol{x}_i^t)$ in line 8 of Algorithm 1. When all local model $\boldsymbol{x}_i^t, \boldsymbol{x}_i^\tau$ and the average model $\hat{\boldsymbol{x}}^t$ converge to the local minimum $\boldsymbol{x}^*$, it holds that

$$
\begin{aligned}
\boldsymbol{v}_i^t &= \nabla f_i(\boldsymbol{x}_i^t) - \frac{1}{k}\sum_{\tau=t''}^{t'-1}\nabla f_i(\boldsymbol{x}_i^\tau) + \frac{1}{Nk}\sum_{\tau=t''}^{t'-1}\sum_{j=1}^{N}\nabla f_j(\boldsymbol{x}_j^\tau) \\
&\to \nabla f_i(\boldsymbol{x}^*) - \frac{1}{k}\sum_{\tau=t''}^{t'-1}\nabla f_i(\boldsymbol{x}^*) + \frac{1}{Nk}\sum_{\tau=t''}^{t'-1}\sum_{j=1}^{N}\nabla f_j(\boldsymbol{x}^*) \\
&\to \frac{1}{Nk}\sum_{\tau=t''}^{t'-1}\sum_{j=1}^{N}\nabla f_j(\boldsymbol{x}^*) \to \nabla f(\boldsymbol{x}^*) \to 0.
\end{aligned}
\tag{11}
$$

Therefore, $\boldsymbol{v}_i^t$ can converge to zero when the variance within each worker is zero, which helps VRL-SGD converge faster. On the other hand, the gradient $\nabla f_i(\boldsymbol{x}_i^t, \xi_i^t)$ in Local SGD cannot converge to zero, which prevents the local model $\boldsymbol{x}_i^\tau$ from converging to the local minimum $\boldsymbol{x}^*$, so it is hard to converge for Local SGD. In summary, that is why VRL-SGD performs better than Local SGD for the *non-identical case*, where the gradient variance among workers is not zero.

## 5 THEORETICAL ANALYSIS

In this section, we provide a theoretical analysis of VRL-SGD. We bound the expected squared gradient norm of the average model, which is the commonly used metric to prove the convergence rate for non-convex problems (Ghadimi & Lan, 2013; Tang et al., 2018; Yu et al., 2019a).

**Theorem 5.1** *Under Assumption 1, if the learning rate satisfies $\gamma \leq \frac{1}{2L}$ and $72k^2\gamma^2L^2 \leq 1$, we have the following convergence result for VRL-SGD in Algorithm 1:*

$$
\frac{1}{T}\sum_{t=0}^{T-1}\mathbb{E}\|\nabla f(\hat{\boldsymbol{x}}^t)\|^2 \leq \frac{3(f(\hat{\boldsymbol{x}}^0) - f^*)}{T\gamma} + \frac{3\gamma L\sigma^2}{2N} + 56k\gamma^2\sigma^2L^2 + \frac{12\gamma^2L^2C}{T},
$$

*where $C$ is defined as*

$$
C = \frac{1}{N}\sum_{t=0}^{k-1}\sum_{i=1}^{N}\left\|\sum_{\tau=0}^{t-1}(\nabla f_i(\hat{\boldsymbol{x}}^\tau) - \nabla f(\hat{\boldsymbol{x}}^\tau))\right\|^2.
\tag{12}
$$

The proof of Theorem 5.1 is given in Appendix C. Note that $C$ will be 0 if $k = 1$ according to equation (12). It is consistent with the fact that VRL-SGD when $k = 1$ is equivalent to S-SGD, where the convergence of S-SGD is not related to the variance among workers.

By setting a suitable learning rate $\gamma$, we have the following corollary.

**Corollary 5.2** *Under Assumption 1, when the learning rate is set as $\gamma = \frac{\sqrt{N}}{\sigma\sqrt{T}}$, the communication period is set as $k = O(T^{\frac{1}{2}}/N^{\frac{3}{2}})$ and the total number of iterations satisfies $T \geq \frac{72N^3L^2k^2}{\sigma^2}$, we have the following convergence result for Algorithm 1:*

$$
\frac{1}{T}\sum_{t=0}^{T-1}\mathbb{E}\left\|\nabla f(\hat{\boldsymbol{x}}^t)\right\| \leq \frac{3\sigma(f(\hat{\boldsymbol{x}}^0) - f^* + 3L)}{\sqrt{NT}} + \frac{12NC}{\sigma^2T^2},
$$

*where $C$ is defined in Theorem 5.1.*

The detailed proof of Corollary 5.2 is given in Appendix D.

**Remark 5.3 Warm-up***. We can set the first communication period $k$ to 1 in VRL-SGD, which is VRL-SGD with a warm-up (VRL-SGD-W), then the variable $C$ in Theorem 5.1 and Corollary 5.2 will be 0. Essentially, this is equivalent to conduct one S-SGD update and initialize $\Delta_i = \nabla f_i(\hat{\boldsymbol{x}}^0, \xi_i^0) - \frac{1}{N}\sum_{j=1}^{N}\nabla f_j(\hat{\boldsymbol{x}}^0, \xi_j^0)$. Therefore, the convergence result is not related to the extent of non-iid. We conduct additional experiments to verify this conclusion in Appendix E.*

**Remark 5.4 Consistent with** $D^2$. *In* $D^2$ *(Tang et al., 2018), the convergence rate is* $O(\frac{1}{\sqrt{NT}} + \frac{\zeta_0^2}{T+\sigma^2T^2})$, *where* $\zeta_0^2$ *represents the extent of non-iid in the first iteration. While the convergence rate in* VRL-SGD *is* $O(\frac{1}{\sqrt{NT}} + \frac{C}{\sigma^2T^2})$, *where* $C$ *is similar to* $\zeta_0^2$. *However, we can reduce the dependence on* $C$ *by a warm-up, which leads to a tighter convergence rate.*

**Remark 5.5 Linear Speedup**. *For non-convex optimization, if there are* $N$ *workers training a model collaboratively, according to Corollary 5.2,* VRL-SGD *converges at the rate* $O(1/\sqrt{NT})$, *which is consistent with S-SGD and Local SGD. To achieve* $\epsilon$-*optimal solutuioin,* $O(\frac{1}{N\epsilon^2})$ *iterations are needed. Thus,* VRL-SGD *has a linear iteration speedup with respect to the number of workers.*

**Remark 5.6 Communication Complexity**. *By Corollary 5.2, to achieve the convergence rate* $O(1/\sqrt{NT})$, *the number of iterations* $T$ *needs to satisfy* $T \geq O(N^3k^2)$, *which requires the communication period* $k \leq O(T^{\frac{1}{2}}/N^{\frac{3}{2}})$. *Consequently, by setting* $k = O(T^{\frac{1}{2}}/N^{\frac{3}{2}})$, VRL-SGD *can reduce communication complexity by a factor* $k$. *However, for the* non-identical case, *previous algorithms based on Local SGD can only reduce communication complexity by a factor* $O(T^{\frac{1}{4}}/N^{\frac{3}{4}})$.

**Remark 5.7 Mini-batch VRL-SGD**. *Although we consider only a single stochastic gradient in each worker so far,* VRL-SGD *can calculate mini-batch gradients with size* $b$ *in line 8 of Algorithm 1. It reduces the variance* $\sigma^2$ *within each worker by a factor* $b$, *thus* VRL-SGD *can converge at the rate* $O(1/\sqrt{bNT})$ *by setting the learning rate* $\gamma = \frac{\sqrt{bN}}{\sigma\sqrt{T}}$.

## 6 EXPERIMENTS

### 6.1 EXPERIMENTAL SETTINGS

**Experimental Environment**  We implement algorithms with Pytorch 1.1 (Paszke et al., 2017). And we use a machine with 8 Nvidia Geforce GTX 1080Ti GPUs, 2 Xeon(R) E5-2620 cores and 256 GB RAM Memory. Each GPU is regarded as one worker in experiments.

**Baselines**  We compare the proposed algorithm VRL-SGD with Local SGD (Stich, 2019), EASGD (Zhang et al., 2015) and S-SGD (Ghadimi & Lan, 2013).

**Data Partitioning**  To validate the effectiveness of VRL-SGD in various scenarios, we consider two cases: the *non-identical case* and the *identical case*. In the *non-identical case*, each worker can only access a subset of data. For example, when 5 workers are used to train a model on 10 classes of data, each worker can only access to two classes of data. In the *identical case*, we allow each worker to access all data.

**Datasets and Models**  We consider three typical tasks: (1) LeNet (El-Sawy et al., 2016) on MNIST (LeCun, 1998); (2) TextCNN (Kim, 2014) on DBPedia (Lehmann et al., 2015); (3) transfer learning on tiny ImageNet [1], which is a subset of the ImageNet dataset (Deng et al., 2009). When training TextCNN on DBPedia, we retain the first 50 words and use a GloVe (Pennington et al., 2014) pre-trained model to extract 50 features for word representation. In transfer learning, we use an Inception V3 (Szegedy et al., 2016) pre-trained model as the feature extractor to extract 2,048 features for each image. Then we train a multilayer perceptron with one fully-connected hidden layer of 1,024 nodes, 200 output nodes, and relu activation. All datasets are summarized in Table 2. A lot of deep learning models use batch normalization (Ioffe & Szegedy, 2015), which assumes that the mini-batches are sampled from the same distribution. Applying batch normalization directly to the *non-identical case* may lead to some other issues, which is beyond the scope of this paper.

**Hyper-parameters**  For the above three different tasks, we set the weight decay to be $10^{-4}$. And we initialize model weights by performing 2 epoch SGD iterations in all experiments. Other detailed hyper-parameters can be found in Table 2.

---

[1]The tiny ImageNet dataset can be downloaded from `https://tiny-imagenet.herokuapp.com`.

Table 2: Parameters used in experiments and a summary of datasets. $N$ denotes the number of workers, $b$ denotes batch size on each worker, $\gamma$ is the learning rate, $k$ is the communication period, $n$ represents the number of data samples and $m$ represents the number of data categories.

| Model | $N$ | $b$ | $\gamma$ | $k$ | Dataset | $n$ | $m$ |
|---|---|---|---|---|---|---|---|
| LeNet | 8 | 32 | 0.005 | 20 | MNIST | 60,000 | 10 |
| TextCNN | 8 | 64 | 0.01 | 50 | DBPedia | 560,000 | 14 |
| Transfer Learning | 8 | 32 | 0.025 | 20 | Tiny ImageNet | 100,000 | 200 |

**Metrics**    In this paper, we mainly focus on the convergence rate of different algorithms. Local SGD has a more superior training speed performance than S-SGD, which has been empirically observed in various machine learning tasks (Povey et al., 2014; Su & Chen, 2015). Besides, VRL-SGD has only a minor change over Local SGD. So VRL-SGD and Local SGD have the same training time in one epoch and both of them have a faster training speed compared with S-SGD. VRL-SGD and EASGD would have the same communication complexity under the same period $k$. Therefore, we compare only the convergence rate (the training loss with regard to epochs) of different algorithms.

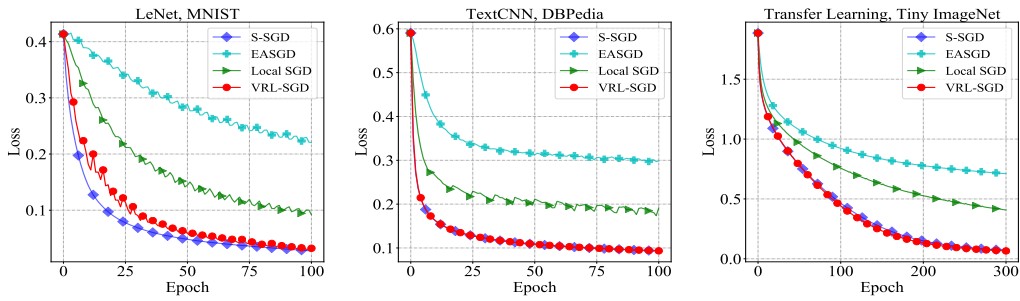

Figure 1: Epoch loss for the *non-identical case*. VRL-SGD converges as fast as S-SGD, and Local SGD, EASGD converge slowly or even cannot converge.

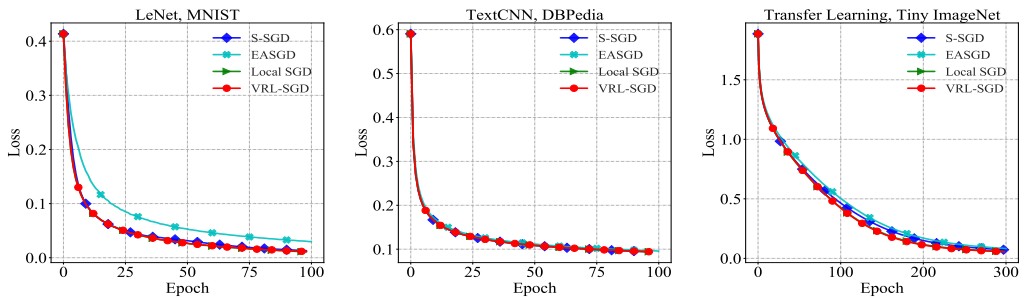

Figure 2: Epoch loss for the *identical case*. All of the algorithms have a similar convergence rate.

## 6.2    NON-IDENTICAL CASE

This paper seeks to address the problem of poor convergence for Local SGD when the variance among workers is high. Therefore, we focus on comparing the convergence rate of all algorithms in the *non-identical case*, where the data variance among workers is maximized.

We choose three classical tasks: image classification, text classification, and transfer learning. Figure 1 shows the training loss with regard to epochs on the three tasks. The results are indicative of the strength of VRL-SGD in the *non-identical case*. Local SGD converges slowly compared with S-SGD when the communication period $k$ is relatively large, while VRL-SGD enjoys the same convergence rate as that of S-SGD. This is consistent with theoretical analysis that VRL-SGD has

a better communication period bound compared to Local SGD. When the variance among workers is not zero, Local SGD requires that $T$ is greater than $O(N^3 k^4)$ to achieve a *linear iteration speedup*. Thus Local SGD loses this property if $k$ is larger than $O(T^{\frac{1}{4}}/N^{\frac{3}{4}})$. However, benefiting from eliminating the dependency on the gradient variance among workers, VRL-SGD can attain a better communication period bound $O(T^{\frac{1}{2}}/N^{\frac{3}{2}})$ than Local SGD as shown in Corollary 5.2. Therefore, under the same communication period, VRL-SGD can achieve a *linear iteration speedup* and converges much faster than Local SGD. To maintain the same convergence rate, Local SGD needs to set a smaller communication period, which will result in higher communication cost. EASGD converges the worst under the same communication period in the *non-identical case*.

There are more experimental results to analyze the influence of parameter $k$ in Appendix F.

### 6.3 IDENTICAL CASE

In addition to the above extreme case, we also validate the effectiveness of VRL-SGD in the *identical case*. As shown in Figure 2, all algorithms have a similar convergence rate. VRL-SGD, EASGD and Local SGD converge as fast as S-SGD when workers can observe unbiased stochastic gradients.

## 7 CONCLUSION & FUTURE WORK

In this paper, we propose a novel distributed algorithm VRL-SGD for accelerating the training of machine learning models. VRL-SGD incorporates the variance reduction technique into Local SGD to further reduce the communication complexity. We theoretically prove that VRL-SGD can achieve a *linear iteration speedup* for nonconvex functions with the optimal communication complexity $O(T^{\frac{1}{2}} N^{\frac{3}{2}})$ whether each worker accesses identical data or not. Experimental results verify the effectiveness of VRL-SGD, where VRL-SGD is significantly better than traditional Local SGD for the *non-identical case* and enjoys the same convergence rate as that of Local SGD.

In the future, we will consider the deep learning models with batch normalization layers, which may lead to an unstable convergence in the *non-identical case*.

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

# A  PROOF OF PARTIALLY ACCUMULATED LOCAL GRADIENTS

In this section, we present Lemma 1 and Lemma 2 to bound the partially accumulated local gradients, which are defined as

$$
\boldsymbol{v}_i^t = \begin{cases} \nabla f(\boldsymbol{x}_i^t, \xi_i^t), & t < k \\ \nabla f(\boldsymbol{x}_i^t, \xi_i^t) + \frac{1}{k} \sum_{\tau'=t''}^{t'-1} (\frac{1}{N} \sum_{j=1}^{N} \nabla f_j(\boldsymbol{x}_j^{\tau'}, \xi_j^{\tau'}) - \nabla f_i(\boldsymbol{x}_i^{\tau'}, \xi_i^{\tau'})). & t \geq k \end{cases} \tag{13}
$$

**Lemma 1** *Under Assumption 1, we have the following inequality for $t \geq k$*

$$
\frac{1}{N} \sum_{i=1}^{N} \mathbb{E} \| \sum_{\tau=t'}^{t-1} \boldsymbol{v}_i^{\tau} \|^2 \leq \frac{12L^2}{N} \sum_{i=1}^{N} \left( k \sum_{\tau=t'}^{t-1} \mathbb{E} \| \boldsymbol{x}_i^{\tau} - \hat{\boldsymbol{x}}^{\tau} \|^2 + 2 \sum_{\tau=t'}^{t-1} \sum_{\tau'=t''}^{t'-1} \mathbb{E} \| \hat{\boldsymbol{x}}^{\tau} - \hat{\boldsymbol{x}}^{\tau'} \|^2 + 2k \sum_{\tau'=t''}^{t'-1} \mathbb{E} \| \hat{\boldsymbol{x}}^{\tau'} - \boldsymbol{x}_i^{\tau'} \|^2 \right)
$$

$$
+ 12k \sum_{\tau'=t''}^{t'-1} \| \nabla f(\hat{\boldsymbol{x}}^{\tau'}) \|^2 + 18k\sigma^2. \tag{14}
$$

*Proof.* By the definition of $\boldsymbol{v}_i^t$ in (13), we have

$$
\frac{1}{N} \sum_{i=1}^{N} \mathbb{E} \left\| \sum_{\tau=t'}^{t-1} \boldsymbol{v}_i^{\tau} \right\|^2
$$

$$
= \frac{1}{N} \sum_{i=1}^{N} \mathbb{E} \left\| \sum_{\tau=t'}^{t-1} \left( \nabla f_i(\boldsymbol{x}_i^{\tau}, \xi_i^{\tau}) + \frac{1}{k} \sum_{\tau'=t''}^{t'-1} \left( \frac{1}{N} \sum_{j=1}^{N} \nabla f_j(\boldsymbol{x}_j^{\tau'}, \xi_j^{\tau'}) - \nabla f_i(\boldsymbol{x}_i^{\tau'}, \xi_i^{\tau'}) \right) \right) \right\|^2
$$

$$
= \frac{1}{N} \sum_{i=1}^{N} \mathbb{E} \left\| \sum_{\tau=t'}^{t-1} \left( \nabla f_i(\boldsymbol{x}_i^{\tau}, \xi_i^{\tau}) - \nabla f_i(\boldsymbol{x}_i^{\tau}) + \frac{1}{k} \sum_{\tau'=t''}^{t'-1} \left( \frac{1}{N} \sum_{j=1}^{N} \left( \nabla f_j(\boldsymbol{x}_j^{\tau'}, \xi_j^{\tau'}) - \nabla f_j(\boldsymbol{x}_j^{\tau'}) \right) \right. \right. \right.
$$

$$
\left. \left. \left. + \nabla f_i(\boldsymbol{x}_i^{\tau'}) - \nabla f_i(\boldsymbol{x}_i^{\tau'}, \xi_i^{\tau'}) \right) + \nabla f_i(\boldsymbol{x}_i^{\tau}) + \frac{1}{k} \sum_{\tau'=t''}^{t'-1} \left( \frac{1}{N} \sum_{j=1}^{N} \nabla f_j(\boldsymbol{x}_j^{\tau'}) - \nabla f_i(\boldsymbol{x}_i^{\tau'}) \right) \right) \right\|^2
$$

$$
\leq \frac{2}{N} \sum_{i=1}^{N} \underbrace{\mathbb{E} \left\| \sum_{\tau=t'}^{t-1} \left( \nabla f_i(\boldsymbol{x}_i^{\tau}, \xi_i^{\tau}) - \nabla f_i(\boldsymbol{x}_i^{\tau}) + \frac{1}{k} \sum_{\tau'=t''}^{t'-1} \left( \frac{1}{N} \sum_{j=1}^{N} \left( \nabla f_j(\boldsymbol{x}_j^{\tau'}, \xi_j^{\tau'}) - \nabla f_j(\boldsymbol{x}_j^{\tau'}) \right) \right. \right. \right.}
$$

$$
\underbrace{\left. \left. \left. + \nabla f_i(\boldsymbol{x}_i^{\tau'}) - \nabla f_i(\boldsymbol{x}_i^{\tau'}, \xi_i^{\tau'}) \right) \right) \right\|^2}_{T_1}
$$

$$
+ \frac{2}{N} \sum_{i=1}^{N} \underbrace{\mathbb{E} \left\| \sum_{\tau=t'}^{t-1} \left( \nabla f_i(\boldsymbol{x}_i^{\tau}) + \frac{1}{k} \sum_{\tau'=t''}^{t'-1} \left( \frac{1}{N} \sum_{j=1}^{N} \nabla f_j(\boldsymbol{x}_j^{\tau'}) - \nabla f_i(\boldsymbol{x}_i^{\tau'}) \right) \right) \right\|^2}_{T_2},
$$

$$\tag{15}$$

where the inequality follows from Cauchy's inequality. We next bound $T_1$ as

$$
T_1 \leq 3 \underbrace{\mathbb{E} \left\| \sum_{\tau=t'}^{t-1} (\nabla f_i(\boldsymbol{x}_i^{\tau}, \xi_i^{\tau}) - \nabla f_i(\boldsymbol{x}_i^{\tau})) \right\|^2}_{T_3} + 3 \underbrace{\mathbb{E} \left\| \frac{(t-t')}{k} \sum_{\tau'=t''}^{t'-1} \left( \nabla f_i(\boldsymbol{x}_i^{\tau'}) - \nabla f_i(\boldsymbol{x}_i^{\tau'}, \xi_i^{\tau'}) \right) \right\|^2}_{T_4}
$$

$$
+ 3 \underbrace{\mathbb{E} \left\| \frac{(t-t')}{k} \sum_{\tau'=t''}^{t'-1} \frac{1}{N} \sum_{j=1}^{N} \left( \nabla f_j(\boldsymbol{x}_j^{\tau'}, \xi_j^{\tau'}) - \nabla f_j(\boldsymbol{x}_j^{\tau'}) \right) \right\|^2}_{T_5}. \tag{16}
$$

Because $\xi_i^t$'s are independent at different time and workers, and the variance of stochastic gradient in each worker is bounded by $\sigma^2$, we can bound $T_3, T_4$ and $T_5$ as

$$
\begin{aligned}
T_3 &= \sum_{\tau=t'}^{t-1} \mathbb{E}\left\|\nabla f_i(\boldsymbol{x}_i^\tau, \xi_i^\tau) - \nabla f_i(\boldsymbol{x}_i^\tau)\right\|^2 \\
&\quad + 2\sum_{t'\leq\tau_1<\tau_2\leq t-1} \mathbb{E}\left\langle \nabla f_i(\boldsymbol{x}_i^{\tau_1}, \xi_i^{\tau_1}) - \nabla f_i(\boldsymbol{x}_i^{\tau_1}), \nabla f_i(\boldsymbol{x}_i^{\tau_2}, \xi_i^{\tau_2}) - \nabla f_i(\boldsymbol{x}_i^{\tau_2})\right\rangle \\
&= \sum_{\tau=t'}^{t-1} \mathbb{E}\left\|\nabla f_i(\boldsymbol{x}_i^\tau, \xi_i^\tau) - \nabla f_i(\boldsymbol{x}_i^\tau)\right\|^2 \\
&\leq (t-t')\sigma^2 \\
&\leq k\sigma^2, &(17)\\
T_4 &= \frac{(t-t')^2}{k^2}\left(\sum_{\tau'=t''}^{t'-1} \mathbb{E}\left\|\nabla f_i(\boldsymbol{x}_i^{\tau'}) - \nabla f_i(\boldsymbol{x}_i^{\tau'}, \xi_i^{\tau'})\right\|^2\right.\\
&\quad \left.+ 2\sum_{t''\leq\tau_1'<\tau_2'\leq t'-1} \mathbb{E}\left\langle \nabla f_i(\boldsymbol{x}_i^{\tau_1'}) - \nabla f_i(\boldsymbol{x}_i^{\tau_1'}, \xi_i^{\tau_1'}), \nabla f_i(\boldsymbol{x}_i^{\tau_2'}) - \nabla f_i(\boldsymbol{x}_i^{\tau_2'}, \xi_i^{\tau_2'})\right\rangle\right) \\
&= \frac{(t-t')^2}{k^2}\sum_{\tau'=t''}^{t'-1} \mathbb{E}\left\|\nabla f_i(\boldsymbol{x}_i^{\tau'}) - \nabla f_i(\boldsymbol{x}_i^{\tau'}, \xi_i^{\tau'})\right\|^2 \\
&\leq k\sigma^2, &(18)\\
T_5 &= \frac{(t-t')^2}{N^2 k^2}\mathbb{E}\left\|\sum_{\tau'=t''}^{t'-1}\sum_{j=1}^{N}\left(\nabla f_j(\boldsymbol{x}_j^{\tau'}, \xi_j^{\tau'}) - \nabla f_j(\boldsymbol{x}_j^{\tau'})\right)\right\|^2 \\
&= \frac{(t-t')^2}{N^2 k^2}\left(\sum_{\tau'=t''}^{t'-1} \mathbb{E}\left\|\sum_{j=1}^{N}\left(\nabla f_j(\boldsymbol{x}_j^{\tau'}, \xi_j^{\tau'}) - \nabla f_j(\boldsymbol{x}_j^{\tau'})\right)\right\|^2\right.\\
&\quad \left.+ 2\sum_{t''\leq\tau_1'<\tau_2'\leq t'-1} \mathbb{E}\left\langle \sum_{j=1}^{N}\left(\nabla f_j(\boldsymbol{x}_j^{\tau_1'}, \xi_j^{\tau_1'}) - \nabla f_j(\boldsymbol{x}_j^{\tau_1'})\right), \sum_{j=1}^{N}\left(\nabla f_j(\boldsymbol{x}_j^{\tau_2'}, \xi_j^{\tau_2'}) - \nabla f_j(\boldsymbol{x}_j^{\tau_2'})\right)\right\rangle\right) \\
&= \frac{(t-t')^2}{N^2 k^2}\sum_{\tau'=t''}^{t'-1} \mathbb{E}\left\|\sum_{j=1}^{N}\left(\nabla f_j(\boldsymbol{x}_j^{\tau'}, \xi_j^{\tau'}) - \nabla f_j(\boldsymbol{x}_j^{\tau'})\right)\right\|^2 \\
&= \frac{(t-t')^2}{N^2 k^2}\sum_{\tau'=t''}^{t'-1}\left(\sum_{j=1}^{N}\mathbb{E}\left\|\nabla f_j(\boldsymbol{x}_j^{\tau'}, \xi_j^{\tau'}) - \nabla f_j(\boldsymbol{x}_j^{\tau'})\right\|^2\right.\\
&\quad \left.+ 2\sum_{1\leq j_1<j_2\leq N}\left\langle \nabla f_{j_1}(\boldsymbol{x}_{j_1}^{\tau'}, \xi_{j_1}^{\tau'}) - \nabla f_{j_1}(\boldsymbol{x}_{j_1}^{\tau'}), \nabla f_{j_2}(\boldsymbol{x}_{j_2}^{\tau'}, \xi_{j_2}^{\tau'}) - \nabla f_{j_2}(\boldsymbol{x}_{j_2}^{\tau'})\right\rangle\right) \\
&= \frac{(t-t')^2}{N^2 k^2}\sum_{\tau'=t''}^{t'-1}\sum_{j=1}^{N}\mathbb{E}\left\|\nabla f_j(\boldsymbol{x}_j^{\tau'}, \xi_j^{\tau'}) - \nabla f_j(\boldsymbol{x}_j^{\tau'})\right\|^2 \\
&\leq \frac{k\sigma^2}{N}. &(19)
\end{aligned}
$$

Substituting (17), (18) and (19) into (16), we have

$$T_1 \leq 3(T_3 + T_4 + T_5) \leq 9k\sigma^2. \tag{20}$$

We next bound $T_2$ as

$$
\begin{aligned}
T_2 &= \mathbb{E}\left\|\sum_{\tau=t'}^{t-1}\left(\nabla f_i(\boldsymbol{x}_i^\tau) + \frac{1}{k}\sum_{\tau'=t''}^{t'-1}\left(\frac{1}{N}\sum_{j=1}^N \nabla f_j(\boldsymbol{x}_j^{\tau'}) - \nabla f_i(\boldsymbol{x}_i^{\tau'})\right)\right)\right\|^2 \\
&= \mathbb{E}\left\|\sum_{\tau=t'}^{t-1}\left(\nabla f_i(\boldsymbol{x}_i^\tau) - \nabla f_i(\hat{\boldsymbol{x}}^\tau) + \nabla f_i(\hat{\boldsymbol{x}}^\tau) - \frac{1}{k}\sum_{\tau'=t''}^{t'-1}\nabla f_i(\hat{\boldsymbol{x}}^{\tau'})\right.\right. \\
&\quad\left.\left. + \frac{1}{k}\sum_{\tau'=t''}^{t'-1}\left(\nabla f_i(\hat{\boldsymbol{x}}^{\tau'}) - \nabla f_i(\boldsymbol{x}_i^{\tau'})\right) + \frac{1}{Nk}\sum_{\tau'=t''}^{t'-1}\sum_{j=1}^N\left(\nabla f_j(\boldsymbol{x}_j^{\tau'}) - \nabla f_j(\hat{\boldsymbol{x}}^{\tau'})\right)\right.\right. \\
&\quad\left.\left. + \frac{1}{k}\sum_{\tau'=t''}^{t'-1}\left(\nabla f(\hat{\boldsymbol{x}}^{\tau'}) - \nabla f(\hat{\boldsymbol{x}}^\tau)\right) + \nabla f(\hat{\boldsymbol{x}}^\tau)\right)\right\|^2 \\
&\leq 6\left(\mathbb{E}\left\|\sum_{\tau=t'}^{t-1}(\nabla f_i(\boldsymbol{x}_i^\tau) - \nabla f_i(\hat{\boldsymbol{x}}^\tau))\right\|^2 + \mathbb{E}\left\|\sum_{\tau=t'}^{t-1}\left(\nabla f_i(\hat{\boldsymbol{x}}^\tau) - \frac{1}{k}\sum_{\tau'=t''}^{t'-1}\nabla f_i(\hat{\boldsymbol{x}}^{\tau'})\right)\right\|^2\right. \\
&\quad + \mathbb{E}\left\|\frac{1}{k}\sum_{\tau=t'}^{t-1}\sum_{\tau'=t''}^{t'-1}\left(\nabla f_i(\hat{\boldsymbol{x}}^{\tau'}) - \nabla f_i(\boldsymbol{x}_i^{\tau'})\right)\right\|^2 + \mathbb{E}\left\|\frac{1}{Nk}\sum_{\tau=t'}^{t-1}\sum_{\tau'=t''}^{t'-1}\sum_{j=1}^N\left(\nabla f_j(\boldsymbol{x}_j^{\tau'}) - \nabla f_j(\hat{\boldsymbol{x}}^{\tau'})\right)\right\|^2 \\
&\quad\left. + \mathbb{E}\left\|\frac{1}{k}\sum_{\tau=t'}^{t-1}\sum_{\tau'=t''}^{t'-1}\left(\nabla f(\hat{\boldsymbol{x}}^{\tau'}) - \nabla f(\hat{\boldsymbol{x}}^\tau)\right)\right\|^2 + \mathbb{E}\left\|\sum_{\tau=t'}^{t-1}\nabla f(\hat{\boldsymbol{x}}^\tau)\right\|^2\right) \\
&\leq 6(t-t')\sum_{\tau=t'}^{t-1}\left(\mathbb{E}\|\nabla f_i(\boldsymbol{x}_i^\tau) - \nabla f_i(\hat{\boldsymbol{x}}^\tau)\|^2 + \mathbb{E}\left\|\nabla f_i(\hat{\boldsymbol{x}}^\tau) - \frac{1}{k}\sum_{\tau'=t''}^{t'-1}\nabla f_i(\hat{\boldsymbol{x}}^{\tau'})\right\|^2\right. \\
&\quad + \mathbb{E}\left\|\frac{1}{k}\sum_{\tau'=t''}^{t'-1}\left(\nabla f_i(\hat{\boldsymbol{x}}^{\tau'}) - \nabla f_i(\boldsymbol{x}_i^{\tau'})\right)\right\|^2 + \mathbb{E}\left\|\frac{1}{Nk}\sum_{\tau'=t''}^{t'-1}\sum_{j=1}^N\left(\nabla f_j(\boldsymbol{x}_j^{\tau'}) - \nabla f_j(\hat{\boldsymbol{x}}^{\tau'})\right)\right\|^2 \\
&\quad\left. + \mathbb{E}\left\|\frac{1}{k}\sum_{\tau=t'}^{t-1}\left(\nabla f(\hat{\boldsymbol{x}}^{\tau'}) - \nabla f(\hat{\boldsymbol{x}}^\tau)\right)\right\|^2 + \mathbb{E}\|\nabla f(\hat{\boldsymbol{x}}^\tau)\|^2\right) \\
&\leq 6(t-t')\sum_{\tau=t'}^{t-1}\left(\mathbb{E}\|\nabla f_i(\boldsymbol{x}_i^\tau) - \nabla f_i(\hat{\boldsymbol{x}}^\tau)\|^2 + \frac{2}{k}\sum_{\tau'=t''}^{t'-1}\mathbb{E}\left\|\nabla f_i(\hat{\boldsymbol{x}}^\tau) - \nabla f_i(\hat{\boldsymbol{x}}^{\tau'})\right\|^2\right. \\
&\quad\left. + \frac{1}{k}\sum_{\tau'=t''}^{t'-1}\mathbb{E}\left\|\nabla f_i(\hat{\boldsymbol{x}}^{\tau'}) - \nabla f_i(\boldsymbol{x}_i^{\tau'})\right\|^2 + \frac{1}{Nk}\sum_{\tau'=t''}^{t'-1}\sum_{j=1}^N\mathbb{E}\left\|\nabla f_j(\boldsymbol{x}_j^{\tau'}) - \nabla f_j(\hat{\boldsymbol{x}}^{\tau'})\right\|^2 + \mathbb{E}\|\nabla f(\hat{\boldsymbol{x}}^\tau)\|^2\right) \\
&\leq 6(t-t')L^2\sum_{\tau=t'}^{t-1}\left(\mathbb{E}\|\boldsymbol{x}_i^\tau - \hat{\boldsymbol{x}}^\tau\|^2 + \frac{2}{k}\sum_{\tau'=t''}^{t'-1}\mathbb{E}\left\|\hat{\boldsymbol{x}}^\tau - \hat{\boldsymbol{x}}^{\tau'}\right\|^2 + \frac{1}{k}\sum_{\tau'=t''}^{t'-1}\mathbb{E}\left\|\hat{\boldsymbol{x}}^{\tau'} - \boldsymbol{x}_i^{\tau'}\right\|^2\right. \\
&\quad\left. + \frac{1}{Nk}\sum_{\tau'=t''}^{t'-1}\sum_{j=1}^N\mathbb{E}\left\|\boldsymbol{x}_j^{\tau'} - \hat{\boldsymbol{x}}^{\tau'}\right\|^2\right) + 6(t-t')\sum_{\tau=t'}^{t-1}\mathbb{E}\|\nabla f(\hat{\boldsymbol{x}}^\tau)\|^2 \\
&\leq 6L^2\left(k\sum_{\tau=t'}^{t-1}\mathbb{E}\|\boldsymbol{x}_i^\tau - \hat{\boldsymbol{x}}^\tau\|^2 + 2\sum_{\tau=t'}^{t-1}\sum_{\tau'=t''}^{t'-1}\mathbb{E}\left\|\hat{\boldsymbol{x}}^\tau - \hat{\boldsymbol{x}}^{\tau'}\right\|^2 + k\sum_{\tau'=t''}^{t'-1}\mathbb{E}\left\|\hat{\boldsymbol{x}}^{\tau'} - \boldsymbol{x}_i^{\tau'}\right\|^2\right. \\
&\quad\left. + \frac{k}{N}\sum_{\tau'=t''}^{t'-1}\sum_{j=1}^N\mathbb{E}\left\|\boldsymbol{x}_j^{\tau'} - \hat{\boldsymbol{x}}^{\tau'}\right\|^2\right) + 6k\sum_{\tau=t'}^{t-1}\mathbb{E}\|\nabla f(\hat{\boldsymbol{x}}^\tau)\|^2, \quad (21)
\end{aligned}
$$

where the first three inequalities follow from Cauchy's inequality, and the fourth inequality follows from the Lipschitz gradient assumption. According to (21), we have

$$
\frac{2}{N}\sum_{i=1}^{N}T_2 \ \le\ \frac{12L^2}{N}\sum_{i=1}^{N}\left(k\sum_{\tau=t'}^{t-1}\mathbb{E}\,\|\boldsymbol{x}_i^\tau - \hat{\boldsymbol{x}}^\tau\|^2 + 2\sum_{\tau=t'}^{t-1}\sum_{\tau'=t''}^{t'-1}\mathbb{E}\left\|\hat{\boldsymbol{x}}^\tau - \hat{\boldsymbol{x}}^{\tau'}\right\|^2 + 2k\sum_{\tau'=t''}^{t'-1}\mathbb{E}\left\|\hat{\boldsymbol{x}}^{\tau'} - \boldsymbol{x}_i^{\tau'}\right\|^2\right)
$$

$$
+12k\sum_{\tau=t'}^{t-1}\mathbb{E}\,\|\nabla f(\hat{\boldsymbol{x}}^\tau)\|^2\,, \tag{22}
$$

Substituting (20 ), (22) into (15), we obtain Lemma 1.

**Lemma 2** *Under Assumption 1, we have the following inequality for $t < k$,*

$$
\frac{1}{N}\sum_{i=1}^{N}\mathbb{E}\left\|\sum_{\tau=t'}^{t-1}\boldsymbol{v}_i^\tau\right\|^2 \le \frac{4kL^2}{N}\sum_{i=1}^{N}\sum_{\tau=t'}^{t-1}\mathbb{E}\|\boldsymbol{x}_i^\tau - \hat{\boldsymbol{x}}^\tau\|^2 + 4k\sum_{\tau=t'}^{t-1}\|\nabla f(\hat{\boldsymbol{x}}^\tau)\|^2 + 4k\sigma^2
$$

$$
+\frac{4}{N}\sum_{i=1}^{N}\left\|\sum_{\tau=t'}^{t-1}(\nabla f_i(\hat{\boldsymbol{x}}^\tau) - \nabla f(\hat{\boldsymbol{x}}^\tau))\right\|^2. \tag{23}
$$

*Proof.* By the definition of $\boldsymbol{v}_i^t, t < k$ in (13), we have

$$
\frac{1}{N}\sum_{i=1}^{N}\mathbb{E}\left\|\sum_{\tau=t'}^{t-1}\boldsymbol{v}_i^\tau\right\|^2
$$

$$
=\ \frac{1}{N}\sum_{i=1}^{N}\mathbb{E}\left\|\sum_{\tau=t'}^{t-1}\nabla f_i(\boldsymbol{x}_i^\tau, \xi_i^\tau)\right\|^2
$$

$$
=\ \frac{1}{N}\sum_{i=1}^{N}\mathbb{E}\left\|\sum_{\tau=t'}^{t-1}\left((\nabla f_i(\boldsymbol{x}_i{}^\tau, \xi_i^\tau) - \nabla f_i(\boldsymbol{x}_i^\tau)) + (\nabla f_i(\boldsymbol{x}_i^\tau) - \nabla f_i(\hat{\boldsymbol{x}}^\tau))\right.\right.
$$

$$
\left.\left. + (\nabla f_i(\hat{\boldsymbol{x}}^\tau) - \nabla f(\hat{\boldsymbol{x}}^\tau)) + \nabla f(\hat{\boldsymbol{x}}^\tau)\right)\right\|^2
$$

$$
\le\ \frac{4}{N}\sum_{i=1}^{N}\left(\mathbb{E}\left\|\sum_{\tau=t'}^{t-1}(\nabla f_i(\boldsymbol{x}_i{}^\tau, \xi_i^\tau) - \nabla f_i(\boldsymbol{x}_i^\tau))\right\|^2 + \mathbb{E}\left\|\sum_{\tau=t'}^{t-1}(\nabla f_i(\boldsymbol{x}_i^\tau) - \nabla f_i(\hat{\boldsymbol{x}}^\tau))\right\|^2\right.
$$

$$
\left. + \mathbb{E}\left\|\sum_{\tau=t'}^{t-1}(\nabla f_i(\hat{\boldsymbol{x}}^\tau) - \nabla f(\hat{\boldsymbol{x}}^\tau))\right\|^2 + \mathbb{E}\left\|\sum_{\tau=t'}^{t-1}\nabla f(\hat{\boldsymbol{x}}^\tau)\right\|^2\right)
$$

$$
\le\ \frac{4}{N}\sum_{i=1}^{N}\left(k\sigma^2 + k\sum_{\tau=t'}^{t-1}\mathbb{E}\,\|\nabla f_i(\boldsymbol{x}_i^\tau) - \nabla f_i(\hat{\boldsymbol{x}}^\tau)\|^2 + \mathbb{E}\left\|\sum_{\tau=t'}^{t-1}(\nabla f_i(\hat{\boldsymbol{x}}^\tau) - \nabla f(\hat{\boldsymbol{x}}^\tau))\right\|^2\right.
$$

$$
\left. + k\sum_{\tau=t'}^{t-1}\mathbb{E}\,\|\nabla f(\hat{\boldsymbol{x}}^\tau)\|^2\right)
$$

$$
\le\ \frac{4}{N}\sum_{i=1}^{N}\left(k\sigma^2 + kL^2\sum_{\tau=t'}^{t-1}\mathbb{E}\,\|\boldsymbol{x}_i^\tau - \hat{\boldsymbol{x}}^\tau\|^2 + \mathbb{E}\left\|\sum_{\tau=t'}^{t-1}(\nabla f_i(\hat{\boldsymbol{x}}^\tau) - \nabla f(\hat{\boldsymbol{x}}^\tau))\right\|^2\right.
$$

$$
\left. + k\sum_{\tau=t'}^{t-1}\mathbb{E}\,\|\nabla f(\hat{\boldsymbol{x}}^\tau)\|^2\right), \tag{24}
$$

where the second inequalities can be obtained by using (17) again. Rerrangeing the inequality, we obtain Lemma 2.

## B  PROOF OF LEMMA 3

In this section, we introduce Lemma 3, which bounds the difference between the local model $\boldsymbol{x}_i^t$ and the average model $\hat{\boldsymbol{x}}^t$.

**Lemma 3** *Under Lemma 1 and Lemma 2 , when the learning rate $\gamma$ and the communication period $k$ satisfy that $72\gamma^2 k^2 L^2 \leq 1$, we have the following inequality*

$$\frac{1}{N} \sum_{t=0}^{T-1} \sum_{i=1}^{N} \mathbb{E}\|\boldsymbol{x}_i^t - \hat{\boldsymbol{x}}^t\|^2 \quad \leq \quad \frac{12k^2\gamma^2}{1 - 36k^2\gamma^2 L^2} \sum_{t=0}^{T-1} \|\nabla f(\hat{\boldsymbol{x}}^t)\|^2 + \frac{24k\gamma^2 L^2}{1 - 36k^2\gamma^2 L^2} \sum_{t=k}^{T-1} \sum_{\tau'=t''}^{t'-1} \mathbb{E}\left\|\hat{\boldsymbol{x}}^t - \hat{\boldsymbol{x}}^{\tau'}\right\|^2$$

$$+\frac{18k\gamma^2\sigma^2 T}{1 - 36k^2\gamma^2 L^2} + \frac{4\gamma^2 C}{1 - 36k^2\gamma^2 L^2}, \tag{25}$$

*where*

$$C = \frac{1}{N} \sum_{t=0}^{k-1} \sum_{i=1}^{N} \left\|\sum_{\tau=0}^{t-1} (\nabla f_i(\hat{\boldsymbol{x}}^\tau) - \nabla f(\hat{\boldsymbol{x}}^\tau))\right\|^2. \tag{26}$$

*Proof.* According to the updating scheme in Algorithms 1, $\boldsymbol{x}_i^t$ can be represented as

$$\boldsymbol{x}_i^t = \hat{\boldsymbol{x}}^{t'} - \gamma \sum_{\tau=t'}^{t-1} \boldsymbol{v}_i^\tau, \tag{27}$$

On the other hand, by the definition of $\hat{\boldsymbol{x}}^t$, we can represent it as

$$\hat{\boldsymbol{x}}^t = \hat{\boldsymbol{x}}^{t'} - \frac{\gamma}{N} \sum_{i=1}^{N} \sum_{\tau=t'}^{t-1} \boldsymbol{v}_i^\tau \tag{28}$$

Substituting (27) and (28) into the left hand side of (25) , we have

$$\frac{1}{N} \sum_{i=1}^{N} \mathbb{E}\left\|\hat{\boldsymbol{x}}^t - \boldsymbol{x}_i^t\right\|^2$$

$$= \quad \frac{1}{N} \sum_{i=1}^{N} \mathbb{E}\left\|\left(\hat{\boldsymbol{x}}^{t'} - \frac{\gamma}{N} \sum_{\tau=t'}^{t-1} \sum_{j=1}^{N} \boldsymbol{v}_j^\tau\right) - \left(\hat{\boldsymbol{x}}^{t'} - \sum_{\tau=t'}^{t-1} \gamma \boldsymbol{v}_i{}^\tau\right)\right\|^2$$

$$= \quad \frac{1}{N} \sum_{i=1}^{N} \mathbb{E}\left\|\sum_{\tau=t'}^{t-1} \gamma \boldsymbol{v}_i{}^\tau - \frac{\gamma}{N} \sum_{\tau=t'}^{t-1} \sum_{j=1}^{N} \boldsymbol{v}_j^\tau\right\|^2$$

$$= \quad \frac{1}{N} \sum_{i=1}^{N} \mathbb{E}\left\|\sum_{\tau=t'}^{t-1} \gamma \boldsymbol{v}_i{}^\tau\right\|^2 + \frac{1}{N} \sum_{i=1}^{N} \mathbb{E}\left\|\frac{\gamma}{N} \sum_{\tau=t'}^{t-1} \sum_{j=1}^{N} \boldsymbol{v}_j^\tau\right\|^2 - 2 \sum_{i=1}^{N} \frac{1}{N} \mathbb{E}\left\langle \sum_{\tau=t'}^{t-1} \gamma \boldsymbol{v}_i{}^\tau, \frac{\gamma}{N} \sum_{\tau=t'}^{t-1} \sum_{j=1}^{N} \boldsymbol{v}_j^\tau \right\rangle$$

$$= \quad \frac{1}{N} \sum_{i=1}^{N} \mathbb{E}\left\|\sum_{\tau=t'}^{t-1} \gamma \boldsymbol{v}_i{}^\tau\right\|^2 + \mathbb{E}\left\|\sum_{\tau=t'}^{t-1} \gamma \sum_{j=1}^{N} \frac{1}{N} \boldsymbol{v}_j^\tau\right\|^2 - 2\mathbb{E}\left\|\sum_{\tau=t'}^{t-1} \gamma \sum_{j=1}^{N} \frac{1}{N} \boldsymbol{v}_j^\tau\right\|^2$$

$$= \quad \frac{1}{N} \sum_{i=1}^{N} \mathbb{E}\left\|\sum_{\tau=t'}^{t-1} \gamma \boldsymbol{v}_i{}^\tau\right\|^2 - \mathbb{E}\left\|\frac{\gamma}{N} \sum_{\tau=t'}^{t-1} \sum_{j=1}^{N} \nabla f_j(\boldsymbol{x}_i^\tau, \xi_j^\tau)\right\|^2$$

$$\leq \quad \frac{1}{N} \sum_{i=1}^{N} \mathbb{E}\left\|\sum_{\tau=t'}^{t-1} \gamma \boldsymbol{v}_i{}^\tau\right\|^2. \tag{29}$$

According to the result in Lemma 1 and Lemma 2, for $t \geq k$, we have

$$\frac{1}{N} \sum_{i=1}^{N} \mathbb{E}\left\|\hat{\boldsymbol{x}}^t - \boldsymbol{x}_i^t\right\|^2 \quad \leq \quad \frac{12\gamma^2 L^2}{N} \sum_{i=1}^{N} \left(k \sum_{\tau=t'}^{t-1} \mathbb{E}\|\boldsymbol{x}_i^\tau - \hat{\boldsymbol{x}}^\tau\|^2 + 2 \sum_{\tau=t'}^{t-1} \sum_{\tau'=t''}^{t'-1} \mathbb{E}\|\hat{\boldsymbol{x}}^\tau - \hat{\boldsymbol{x}}^{\tau'}\|^2\right.$$

$$\left.+2k \sum_{\tau'=t''}^{t'-1} \mathbb{E}\|\hat{\boldsymbol{x}}^{\tau'} - \boldsymbol{x}_i^{\tau'}\|^2\right) + 12k\gamma^2 \sum_{\tau'=t''}^{t'-1} \|\nabla f(\hat{\boldsymbol{x}}^{\tau'})\|^2 + 18k\gamma^2\sigma^2, \tag{30}$$

and for $t < k$, we have

$$\frac{1}{N} \sum_{i=1}^{N} \mathbb{E}\left\|\hat{\boldsymbol{x}}^t - \boldsymbol{x}_i^t\right\|^2 \quad \leq \quad \frac{4k\gamma^2 L^2}{N} \sum_{i=1}^{N} \sum_{\tau=0}^{t-1} \mathbb{E}\|\boldsymbol{x}_i^\tau - \hat{\boldsymbol{x}}^\tau\|^2 + 4k\gamma^2 \sum_{\tau=0}^{t-1} \|\nabla f(\hat{\boldsymbol{x}}^\tau)\|^2$$

$$+4k\gamma^2\sigma^2 + \frac{4\gamma^2}{N} \sum_{i=1}^{N} \left\|\sum_{\tau=t'}^{t-1} (\nabla f_i(\hat{\boldsymbol{x}}^\tau) - \nabla f(\hat{\boldsymbol{x}}^\tau))\right\|^2 \tag{31}$$

Summing up (30) and (31) from $t = 0$ to $T - 1$, we obtain

$$
\frac{1}{N} \sum_{t=0}^{T-1} \sum_{i=1}^{N} \mathbb{E} \left\| \hat{\boldsymbol{x}}^t - \boldsymbol{x}_i^t \right\|^2
$$

$$
\leq \quad \frac{12\gamma^2 L^2}{N} \sum_{t=k}^{T-1} \sum_{i=1}^{N} \left( k \sum_{\tau=t'}^{t-1} \mathbb{E}\|\boldsymbol{x}_i^\tau - \hat{\boldsymbol{x}}^\tau\|^2 + 2 \sum_{\tau=t'}^{t-1} \sum_{\tau'=t''}^{t'-1} \mathbb{E}\|\hat{\boldsymbol{x}}^\tau - \hat{\boldsymbol{x}}^{\tau'}\|^2 + 2k \sum_{\tau'=t''}^{t'-1} \mathbb{E}\|\hat{\boldsymbol{x}}^{\tau'} - \boldsymbol{x}_i^{\tau'}\|^2 \right)
$$

$$
+ 12k\gamma^2 \sum_{t=k}^{T-1} \sum_{\tau=t'}^{t-1} \|\nabla f(\hat{\boldsymbol{x}}^\tau)\|^2 + 18k\gamma^2\sigma^2 (T-k) + \frac{4k\gamma^2 L^2}{N} \sum_{t=0}^{k-1} \sum_{i=1}^{N} \sum_{\tau=t'}^{t-1} \mathbb{E}\|\boldsymbol{x}_i^\tau - \hat{\boldsymbol{x}}^\tau\|^2
$$

$$
+ 4k\gamma^2 \sum_{t=0}^{k-1} \sum_{\tau=t'}^{t-1} \|\nabla f(\hat{\boldsymbol{x}}^\tau)\|^2 + 4k^2\gamma^2\sigma^2 + \frac{4\gamma^2}{N} \sum_{t=0}^{k-1} \sum_{i=1}^{N} \left\| \sum_{\tau=t'}^{t-1} (\nabla f_i(\hat{\boldsymbol{x}}^\tau) - \nabla f(\hat{\boldsymbol{x}}^\tau)) \right\|^2
$$

$$
\leq \quad \frac{12\gamma^2 L^2}{N} \sum_{t=0}^{T-1} \sum_{i=1}^{N} 3k^2 \mathbb{E}\|\boldsymbol{x}_i^t - \hat{\boldsymbol{x}}^t\|^2 + 24\gamma^2 L^2 \sum_{t=k}^{T-1} \sum_{\tau=t'}^{t-1} \sum_{\tau'=t''}^{t'-1} \mathbb{E}\|\hat{\boldsymbol{x}}^\tau - \hat{\boldsymbol{x}}^{\tau'}\|^2
$$

$$
+ 12k^2\gamma^2 \sum_{t=0}^{T-1} \|\nabla f(\hat{\boldsymbol{x}}^t)\|^2 + 18k\gamma^2\sigma^2 T + \frac{4\gamma^2}{N} \sum_{t=0}^{k-1} \sum_{i=1}^{N} \left\| \sum_{\tau=t'}^{t-1} (\nabla f_i(\hat{\boldsymbol{x}}^\tau) - \nabla f(\hat{\boldsymbol{x}}^\tau)) \right\|^2
$$

$$
\leq \quad \frac{36\gamma^2 k^2 L^2}{N} \sum_{t=0}^{T-1} \sum_{i=1}^{N} \mathbb{E}\|\boldsymbol{x}_i^t - \hat{\boldsymbol{x}}^t\|^2 + 24k\gamma^2 L^2 \sum_{t=k}^{T-1} \sum_{\tau'=t''}^{t'-1} \mathbb{E}\|\hat{\boldsymbol{x}}^t - \hat{\boldsymbol{x}}^{\tau'}\|^2
$$

$$
+ 12k^2\gamma^2 \sum_{t=0}^{T-1} \|\nabla f(\hat{\boldsymbol{x}}^t)\|^2 + 18k\gamma^2\sigma^2 T + \frac{4\gamma^2}{N} \sum_{t=0}^{k-1} \sum_{i=1}^{N} \left\| \sum_{\tau=t'}^{t-1} (\nabla f_i(\hat{\boldsymbol{x}}^\tau) - \nabla f(\hat{\boldsymbol{x}}^\tau)) \right\|^2, \tag{32}
$$

where the second and the third inequalities can be obtained by using a simple counting argument. Denote $C = \frac{1}{N} \sum_{t=0}^{k-1} \sum_{i=1}^{N} \left\| \sum_{\tau=0}^{t-1} (\nabla f_i(\hat{\boldsymbol{x}}^\tau) - \nabla f(\hat{\boldsymbol{x}}^\tau)) \right\|^2$. Rerrangeing the inequality, we obtain

$$
(1 - 36k^2\gamma^2 L^2)\frac{1}{N} \sum_{t=0}^{T-1} \sum_{i=1}^{N} \mathbb{E}\|\boldsymbol{x}_i^t - \hat{\boldsymbol{x}}^t\|^2 \quad \leq \quad 12k^2\gamma^2 \sum_{t=0}^{T-1} \|\nabla f(\hat{\boldsymbol{x}}^t)\|^2 + 24k\gamma^2 L^2 \sum_{t=0}^{T-1} \sum_{\tau'=t''}^{t'-1} \mathbb{E}\|\hat{\boldsymbol{x}}^t - \hat{\boldsymbol{x}}^{\tau'}\|^2
$$

$$
+ 18k\gamma^2\sigma^2 T + 4\gamma^2 C. \tag{33}
$$

Dividing $1 - 36k^2\gamma^2 L^2$ on both sides completes the proof.

## C   PROOF OF THEOREM 5.1

In this section, we give the proof of Theorem 5.1.

**Theorem 5.1** *Under Assumption 1, if the learning rate satisfies $\gamma \leq \frac{1}{2L}$ and $72k^2\gamma^2 L^2 \leq 1$, we have the following convergence result for Algorithm 1:*

$$
\frac{1}{T} \sum_{t=0}^{T-1} \mathbb{E}\|\nabla f(\hat{\boldsymbol{x}}^t)\|^2 \leq \frac{3(f(\hat{\boldsymbol{x}}^0) - f^*)}{T\gamma} + \frac{3\gamma L\sigma^2}{2N} + 56k\gamma^2\sigma^2 L^2 + \frac{12\gamma^2 L^2 C}{T}. \tag{34}
$$

*Proof.* Since $f_i(\cdot), i = 1, 2, \cdots, N$ are $L$-smooth, it is easy to verify that $f(\cdot)$ is $L$-smooth. We have

$$
f(\hat{\boldsymbol{x}}_{t+1}) \quad \leq \quad f(\hat{\boldsymbol{x}}^t) + \left\langle \nabla f(\hat{\boldsymbol{x}}^t), \hat{\boldsymbol{x}}^{t+1} - \hat{\boldsymbol{x}}^t \right\rangle + \frac{L}{2} \left\| \hat{\boldsymbol{x}}^{t+1} - \hat{\boldsymbol{x}}^t \right\|^2
$$

$$
= \quad f(\hat{\boldsymbol{x}}^t) - \gamma \left\langle \nabla f(\hat{\boldsymbol{x}}^t), \frac{1}{N} \sum_{i=1}^{N} \boldsymbol{v}_i^t \right\rangle + \frac{L\gamma^2}{2} \left\| \frac{1}{N} \sum_{i=1}^{N} \boldsymbol{v}_i^t \right\|^2
$$

$$
= \quad f(\hat{\boldsymbol{x}}^t) - \gamma \left\langle \nabla f(\hat{\boldsymbol{x}}^t), \frac{1}{N} \sum_{i=1}^{N} \nabla f_i(\boldsymbol{x}_i^t, \xi_i^t) \right\rangle + \frac{L\gamma^2}{2} \left\| \frac{1}{N} \sum_{i=1}^{N} \nabla f_i(\boldsymbol{x}_i^t, \xi_i^t) \right\|^2. \tag{35}
$$

By applying expectation with respect to all the random variables at step $t$ and conditional on the past (denote by $\mathbb{E}_{t|.}$), we have

$$
\begin{aligned}
&\mathbb{E}_{t|.} f(\hat{\boldsymbol{x}}_{t+1}) \\
\leq\ & f(\hat{\boldsymbol{x}}^t) - \gamma \left\langle \nabla f(\hat{\boldsymbol{x}}^t), \frac{1}{N} \sum_{i=1}^{N} \nabla f_i(\boldsymbol{x}_i^t) \right\rangle + \frac{L\gamma^2}{2} \mathbb{E}_{t|.} \left\| \frac{1}{N} \sum_{i=1}^{N} \nabla f_i(\boldsymbol{x}_i^t, \xi_i^t) \right\|^2 \\
=\ & f(\hat{\boldsymbol{x}}^t) - \frac{\gamma}{2} \left( \|\nabla f(\hat{\boldsymbol{x}}^t)\|^2 + \left\| \frac{1}{N} \sum_{i=1}^{N} \nabla f_i(\boldsymbol{x}_i^t) \right\|^2 - \left\| \nabla f(\hat{\boldsymbol{x}}^t) - \frac{1}{N} \sum_{i=1}^{N} \nabla f_i(\boldsymbol{x}_i^t) \right\|^2 \right) \\
& + \frac{L\gamma^2}{2} \mathbb{E}_{t|.} \left\| \frac{1}{N} \sum_{i=1}^{N} \nabla f_i(\boldsymbol{x}_i^t, \xi_i^t) \right\|^2 .
\end{aligned}
\tag{36}
$$

Note that

$$
\begin{aligned}
&\mathbb{E}_{t|.} \left\| \frac{1}{N} \sum_{i=1}^{N} \nabla f_i(\boldsymbol{x}_i^t, \xi_i^t) \right\|^2 \\
=\ & \mathbb{E}_{t|.} \left\| \frac{1}{N} \sum_{i=1}^{N} \nabla f_i(\boldsymbol{x}_i^t, \xi_i^t) - \frac{1}{N} \sum_{i=1}^{N} \nabla f_i(\boldsymbol{x}_i^t) + \frac{1}{N} \sum_{i=1}^{N} \nabla f_i(\boldsymbol{x}_i^t) \right\|^2 \\
=\ & \mathbb{E}_{t|.} \left\| \frac{1}{N} \sum_{i=1}^{N} \nabla f_i(\boldsymbol{x}_i^t, \xi_i^t) - \frac{1}{N} \sum_{i=1}^{N} \nabla f_i(\boldsymbol{x}_i^t) \right\|^2 + \left\| \frac{1}{N} \sum_{i=1}^{N} \nabla f_i(\boldsymbol{x}_i^t) \right\|^2 \\
& + 2\mathbb{E}_{t|.} \left\langle \frac{1}{N} \sum_{i=1}^{N} \nabla f_i(\boldsymbol{x}_i^t, \xi_i^t) - \frac{1}{N} \sum_{i=1}^{N} \nabla f_i(\boldsymbol{x}_i^t), \frac{1}{N} \sum_{i=1}^{N} \nabla f_i(\boldsymbol{x}_i^t) \right\rangle \\
=\ & \mathbb{E}_{t|.} \left\| \frac{1}{N} \sum_{i=1}^{N} \nabla f_i(\boldsymbol{x}_i^t, \xi_i^t) - \frac{1}{N} \sum_{i=1}^{N} \nabla f_i(\boldsymbol{x}_i^t) \right\|^2 + \left\| \frac{1}{N} \sum_{i=1}^{N} \nabla f_i(\boldsymbol{x}_i^t) \right\|^2 ,
\end{aligned}
\tag{37}
$$

where the last equality holds because $\mathbb{E}_{t|.} \left( \frac{1}{N} \sum_{i=1}^{N} \nabla f_i(\boldsymbol{x}_i^t, \xi_i^t) - \frac{1}{N} \sum_{i=1}^{N} \nabla f_i(\boldsymbol{x}_i^t) \right) = 0$, and

$$
\begin{aligned}
&\mathbb{E}_{t|.} \left\| \frac{1}{N} \sum_{i=1}^{N} \nabla f_i(\boldsymbol{x}_i^t, \xi_i^t) - \frac{1}{N} \sum_{i=1}^{N} \nabla f_i(\boldsymbol{x}_i^t) \right\|^2 \\
=\ & \mathbb{E}_{t|.} \frac{1}{N^2} \sum_{i=1}^{N} \left\| \nabla f_i(\boldsymbol{x}_i^t, \xi_i^t) - \nabla f_i(\boldsymbol{x}_i^t) \right\|^2 \\
& + \frac{2}{N^2} \sum_{1 \leq i_1 < i_2 \leq N} \mathbb{E}_{t|.} \left\langle \nabla f_{i_1}(\boldsymbol{x}_{i_1}^t, \xi_{i_1}^t) - \nabla f_{i_1}(\boldsymbol{x}_{i_1}^t), \nabla f_{i_2}(\boldsymbol{x}_{i_2}^t, \xi_{i_2}^t) - \nabla f_{i_2}(\boldsymbol{x}_{i_2}^t) \right\rangle \\
=\ & \mathbb{E}_{t|.} \frac{1}{N^2} \sum_{i=1}^{N} \left\| \nabla f_i(\boldsymbol{x}_i^t, \xi_i^t) - \nabla f_i(\boldsymbol{x}_i^t) \right\|^2 \\
\leq\ & \frac{\sigma^2}{N},
\end{aligned}
\tag{38}
$$

where the second equality holds because the random variables on different workers are independent. Substituting (37) into (36) and applying expectation with respect to all the random variables, we obtain

$$
\begin{aligned}
\mathbb{E} f(\hat{\boldsymbol{x}}_{t+1}) \leq\ & \mathbb{E} f(\hat{\boldsymbol{x}}^t) - \frac{\gamma}{2} \mathbb{E} \|\nabla f(\hat{\boldsymbol{x}}^t)\|^2 - \frac{\gamma}{2}(1 - L\gamma) \mathbb{E} \left\| \frac{1}{N} \sum_{i=1}^{N} \nabla f_i(\boldsymbol{x}_i^t) \right\|^2 \\
& + \frac{\gamma}{2} \mathbb{E} \left\| \nabla f(\hat{\boldsymbol{x}}^t) - \frac{1}{N} \sum_{i=1}^{N} \nabla f_i(\boldsymbol{x}_i^t) \right\|^2 + \frac{\gamma^2 L \sigma^2}{2N} .
\end{aligned}
\tag{39}
$$

We then bound the difference of $\nabla f(\hat{\boldsymbol{x}}^t)$ and $\frac{1}{N}\sum_{i=1}^{N}\nabla f_i(\boldsymbol{x}_i^t)$ as

$$
\begin{aligned}
\mathbb{E}\left\|\nabla f(\hat{\boldsymbol{x}}^t) - \frac{1}{N}\sum_{i=1}^{N}\nabla f_i(\boldsymbol{x}_i^t)\right\|^2 &= \mathbb{E}\left\|\frac{1}{N}\sum_{i=1}^{N}\left(\nabla f_i(\hat{\boldsymbol{x}}^t) - \nabla f_i(\boldsymbol{x}_i^t)\right)\right\|^2 \\
&\leq \frac{1}{N}\sum_{i=1}^{N}\mathbb{E}\left\|\left(\nabla f_i(\hat{\boldsymbol{x}}^t) - \nabla f_i(\boldsymbol{x}_i^t)\right)\right\|^2 \\
&\leq \frac{L^2}{N}\sum_{i=1}^{N}\mathbb{E}\left\|\hat{\boldsymbol{x}}^t - \boldsymbol{x}_i^t\right\|^2,
\end{aligned}
\tag{40}
$$

where the two inequalities follow from Cauchy's inequality and Lipschitz gradient assumption, respectively. Substituting (40) into (39) yields

$$
\begin{aligned}
\mathbb{E}f(\hat{\boldsymbol{x}}_{t+1}) \leq \mathbb{E}f(\hat{\boldsymbol{x}}^t) - \frac{\gamma}{2}\mathbb{E}\|\nabla f(\hat{\boldsymbol{x}}^t)\|^2 - \frac{\gamma}{2}(1-L\gamma)\mathbb{E}\left\|\frac{1}{N}\sum_{i=1}^{N}\nabla f_i(\boldsymbol{x}_i^t)\right\|^2 \\
+ \frac{\gamma L^2}{2N}\sum_{i=1}^{N}\mathbb{E}\|\hat{\boldsymbol{x}}^t - \boldsymbol{x}_i^t\|^2 + \frac{\gamma^2 L\sigma^2}{2N}.
\end{aligned}
\tag{41}
$$

Rearranging the inequality and summing up both sides from $t=0$ to $T-1$, we have

$$
\begin{aligned}
\sum_{t=0}^{T-1}&\left(\frac{\gamma}{2}\mathbb{E}\|\nabla f(\hat{\boldsymbol{x}}^t)\|^2 + \frac{\gamma}{2}(1-L\gamma)\mathbb{E}\left\|\frac{1}{N}\sum_{i=1}^{N}\nabla f_i(\boldsymbol{x}_i^t)\right\|^2\right) \\
\leq\quad & f(\hat{\boldsymbol{x}}_0) - f^* + \frac{\gamma L^2}{2N}\sum_{i=1}^{N}\sum_{t=0}^{T-1}\mathbb{E}\|\hat{\boldsymbol{x}}^t - \boldsymbol{x}_i^t\|^2 + \frac{T\gamma^2 L\sigma^2}{2N}.
\end{aligned}
\tag{42}
$$

Substituting Lemma 3 into (42) and combing $72k^2\gamma^2L^2 \leq 1$, we obtain

$$
\begin{aligned}
\sum_{t=0}^{T-1}&\left(\frac{\gamma}{2}\mathbb{E}\|\nabla f(\hat{\boldsymbol{x}}^t)\|^2 + \frac{\gamma}{2}(1-L\gamma)\mathbb{E}\left\|\frac{1}{N}\sum_{i=1}^{N}\nabla f_i(\boldsymbol{x}_i^t)\right\|^2\right) \\
\leq\quad & f(\hat{\boldsymbol{x}}_0) - f^* + \frac{T\gamma^2 L\sigma^2}{2N} + \frac{6k^2\gamma^3 L^2}{1-36k^2\gamma^2 L^2}\sum_{t=0}^{T-1}\|\nabla f(\hat{\boldsymbol{x}}^t)\|^2 + \frac{9k\gamma^3\sigma^2 L^2 T}{1-36k^2 L^2\gamma^2} \\
& + \frac{2\gamma^3 L^2 C}{1-36k^2\gamma^2 L^2} + \frac{12k\gamma^3 L^4}{1-36k^2\gamma^2 L^2}\sum_{t=k}^{T-1}\sum_{\tau'=t''}^{t'-1}\mathbb{E}\left\|\hat{\boldsymbol{x}}^t - \hat{\boldsymbol{x}}^{\tau'}\right\|^2 \\
\leq\quad & f(\hat{\boldsymbol{x}}_0) - f^* + \frac{T\gamma^2 L\sigma^2}{2N} + 12k^2\gamma^3 L^2\sum_{t=0}^{T-1}\|\nabla f(\hat{\boldsymbol{x}}^t)\|^2 + 18k\gamma^3\sigma^2 L^2 T \\
& + 4\gamma^3 L^2 C + 24k\gamma^3 L^4 \underbrace{\sum_{t=k}^{T-1}\sum_{\tau'=t''}^{t'-1}\mathbb{E}\left\|\hat{\boldsymbol{x}}^t - \hat{\boldsymbol{x}}^{\tau'}\right\|^2}_{T_6}.
\end{aligned}
\tag{43}
$$

Next, we bound $T_6$.

$$
\begin{aligned}
T_6 &= \sum_{t=k}^{T-1} \sum_{\tau'=t''}^{t'-1} \mathbb{E} \left\| \hat{\boldsymbol{x}}^t - \hat{\boldsymbol{x}}^{\tau'} \right\|^2 \\
&= \sum_{t=k}^{T-1} \sum_{\tau'=t''}^{t'-1} \mathbb{E} \left\| \sum_{s=\tau'}^{t-1} \frac{\gamma}{N} \sum_{i=1}^{N} \boldsymbol{v}_i^s \right\|^2 \\
&= \frac{\gamma^2}{N^2} \sum_{t=0}^{T-1} \sum_{\tau'=t''}^{t'-1} \mathbb{E} \left\| \sum_{s=\tau'}^{t-1} \sum_{i=1}^{N} (\nabla f_i(\boldsymbol{x}_i^s, \xi_i^s) - \nabla f_i(\boldsymbol{x}_i^s)) + \sum_{s=\tau'}^{t-1} \sum_{i=1}^{N} \nabla f_i(\boldsymbol{x}_i^s) \right\|^2 \\
&= \frac{\gamma^2}{N^2} \sum_{t=k}^{T-1} \sum_{\tau'=t''}^{t'-1} \left( \mathbb{E} \left\| \sum_{s=\tau'}^{t-1} \sum_{i=1}^{N} (\nabla f_i(\boldsymbol{x}_i^s, \xi_i^s) - \nabla f_i(\boldsymbol{x}_i^s)) \right\|^2 + \mathbb{E} \left\| \sum_{s=\tau'}^{t-1} \sum_{i=1}^{N} \nabla f_i(\boldsymbol{x}_i^s) \right\|^2 \right. \\
&\quad \left. + 2\mathbb{E} \left\langle \sum_{s=\tau'}^{t-1} \sum_{i=1}^{N} (\nabla f_i(\boldsymbol{x}_i^s, \xi_i^s) - \nabla f_i(\boldsymbol{x}_i^s)), \sum_{s=\tau'}^{t-1} \sum_{i=1}^{N} \nabla f_i(\boldsymbol{x}_i^s) \right\rangle \right) \\
&= \frac{\gamma^2}{N^2} \sum_{t=k}^{T-1} \sum_{\tau'=t''}^{t'-1} \underbrace{\mathbb{E} \left\| \sum_{s=\tau'}^{t-1} \sum_{i=1}^{N} (\nabla f_i(\boldsymbol{x}_i^s, \xi_i^s) - \nabla f_i(\boldsymbol{x}_i^s)) \right\|^2}_{T_7} + \frac{\gamma^2}{N^2} \sum_{t=k}^{T-1} \sum_{\tau'=t''}^{t'-1} \mathbb{E} \left\| \sum_{s=\tau'}^{t-1} \sum_{i=1}^{N} \nabla f_i(\boldsymbol{x}_i^s) \right\|^2 .
\end{aligned}
\tag{44}
$$

Since $\xi_i^t$'s are independent, we have

$$
\begin{aligned}
T_7 &= \sum_{s=\tau'}^{t-1} \left( \mathbb{E} \left\| \sum_{i=1}^{N} (\nabla f_i(\boldsymbol{x}_i^s, \xi_i^s) - \nabla f_i(\boldsymbol{x}_i^s)) \right\|^2 \right. \\
&\quad \left. + 2 \sum_{\tau' \le s_1 < s_2 \le t-1} \mathbb{E} \left\langle \sum_{i=1}^{N} (\nabla f_i(\boldsymbol{x}_i^{s_1}, \xi_i^{s_1}) - \nabla f_i(\boldsymbol{x}_i^{s_1})), \sum_{i=1}^{N} (\nabla f_i(\boldsymbol{x}_i^{s_2}, \xi_i^{s_2}) - \nabla f_i(\boldsymbol{x}_i^{s_2})) \right\rangle \right) \\
&= \sum_{s=\tau'}^{t-1} \mathbb{E} \left\| \sum_{i=1}^{N} (\nabla f_i(\boldsymbol{x}_i^s, \xi_i^s) - \nabla f_i(\boldsymbol{x}_i^s)) \right\|^2 \\
&= \sum_{s=\tau'}^{t-1} \left( \sum_{i=1}^{N} \mathbb{E} \| \nabla f_i(\boldsymbol{x}_i^s, \xi_i^s) - \nabla f_i(\boldsymbol{x}_i^s) \|^2 \right. \\
&\quad \left. + 2 \sum_{1 \le i_1 < i_2 \le N} \mathbb{E} \langle \nabla f_{i_1}(\boldsymbol{x}_{i_1}^s, \xi_{i_1}^s) - \nabla f_{i_1}(\boldsymbol{x}_{i_1}^s), \nabla f_{i_2}(\boldsymbol{x}_{i_2}^s, \xi_{i_2}^s) - \nabla f_{i_2}(\boldsymbol{x}_{i_2}^s) \rangle \right) \\
&= \sum_{s=\tau'}^{t-1} \sum_{i=1}^{N} \mathbb{E} \| \nabla f_i(\boldsymbol{x}_i^s, \xi_i^s) - \nabla f_i(\boldsymbol{x}_i^s) \|^2 .
\end{aligned}
\tag{45}
$$

Substituting (45) into (44), we have

$$
\begin{aligned}
T_6 &= \frac{\gamma^2}{N^2} \sum_{t=k}^{T-1} \sum_{\tau'=t''}^{t'-1} \sum_{s=\tau'}^{t-1} \sum_{i=1}^{N} \mathbb{E} \| \nabla f_i(\boldsymbol{x}_i^s, \xi_i^s) - \nabla f_i(\boldsymbol{x}_i^s) \|^2 + \frac{\gamma^2}{N^2} \sum_{t=k}^{T-1} \sum_{\tau'=t''}^{t'-1} \mathbb{E} \left\| \sum_{s=\tau'}^{t-1} \sum_{i=1}^{N} \nabla f_i(\boldsymbol{x}_i^s) \right\|^2 \\
&\le \frac{2k^2 \gamma^2 \sigma^2 T}{N} + \sum_{t=k}^{T-1} \sum_{\tau'=t''}^{t'-1} \mathbb{E} \left\| \frac{\gamma}{N} \sum_{s=\tau'}^{t-1} \sum_{i=1}^{N} \nabla f_i(\boldsymbol{x}_i^s) \right\|^2 ,
\end{aligned}
\tag{46}
$$

where the inequality holds since $t - \tau' \le k \le t - t'' \le 2k$. Substituting (46) into (43), we obtain

$$
\sum_{t=0}^{T-1} \left( \frac{\gamma}{2} \mathbb{E} \| \nabla f(\hat{\boldsymbol{x}}^t) \|^2 + \frac{\gamma}{2} (1 - L\gamma) \mathbb{E} \left\| \frac{1}{N} \sum_{i=1}^{N} \nabla f_i(\boldsymbol{x}_i^t) \right\|^2 \right)
$$

$$
\le \quad f(\hat{\boldsymbol{x}}_0) - f^* + \frac{T\gamma^2 L \sigma^2}{2N} + 12k^2 \gamma^3 L^2 \sum_{t=0}^{T-1} \| \nabla f(\hat{\boldsymbol{x}}^t) \|^2 + 18k\gamma^3 \sigma^2 L^2 T
$$

$$
+ 4\gamma^3 L^2 C + 24k\gamma^3 L^4 \sum_{t=k}^{T-1} \sum_{\tau'=t''}^{t'-1} \mathbb{E} \left\| \hat{\boldsymbol{x}}^t - \hat{\boldsymbol{x}}^{\tau'} \right\|^2
$$

$$
\le \quad f(\hat{\boldsymbol{x}}_0) - f^* + \frac{T\gamma^2 L \sigma^2}{2N} + 12k^2 \gamma^3 L^2 \sum_{t=0}^{T-1} \| \nabla f(\hat{\boldsymbol{x}}^t) \|^2 + 18k\gamma^3 \sigma^2 L^2 T
$$

$$
+ 4\gamma^3 L^2 C + \frac{48k^3 \gamma^5 \sigma^2 L^4 T}{N} + 24k\gamma^3 L^4 \sum_{t=k}^{T-1} \sum_{\tau'=t''}^{t'-1} \left\| \frac{\gamma}{N} \sum_{s=\tau'}^{t-1} \sum_{i=1}^{N} \nabla f_i(\boldsymbol{x}_i^s) \right\|^2. \tag{47}
$$

Rearranging this inequality and dividing both sides by $\frac{T\gamma}{2}$, we get

$$
\frac{1}{T} \sum_{t=0}^{T-1} \left( 1 - 24k^2 \gamma^2 L^2 \right) \mathbb{E} \| \nabla f(\hat{\boldsymbol{x}}^t) \|^2
$$

$$
\le \quad \frac{2(f(\hat{\boldsymbol{x}}^0) - f^*)}{T\gamma} + \frac{\gamma L \sigma^2}{N} + 36k\gamma^2 \sigma^2 L^2 + \frac{8\gamma^2 L^2 C}{T} + \frac{96k^3 \gamma^4 \sigma^2 L^4}{N}
$$

$$
+ \underbrace{\frac{1}{T} \sum_{t=k}^{T-1} 48k\gamma^2 L^4 \sum_{\tau'=t''}^{t'-1} \left\| \frac{\gamma}{N} \sum_{s=\tau'}^{t-1} \sum_{i=1}^{N} \nabla f_i(\boldsymbol{x}_i^s) \right\|^2 - \frac{1}{T} \sum_{t=0}^{T-1} (1 - L\gamma) \mathbb{E} \left\| \frac{1}{N} \sum_{i=1}^{N} \nabla f_i(\boldsymbol{x}_i^t) \right\|^2}_{T_8}. \tag{48}
$$

Then we prove $T_8 \le 0$. If the learnign rate $\gamma$ satisfies $\gamma \le \frac{1}{2L}$, then we have $(1 - L\gamma) \ge \frac{1}{2}$.

$$
T_8 \le \quad \frac{1}{2T} \left( \sum_{t=k}^{T-1} 96k\gamma^4 L^4 \sum_{\tau'=t''}^{t'-1} \left\| \frac{1}{N} \sum_{s=\tau'}^{t-1} \sum_{i=1}^{N} \nabla f_i(\boldsymbol{x}_i^s) \right\|^2 - \sum_{t=0}^{T-1} \mathbb{E} \left\| \frac{1}{N} \sum_{i=1}^{N} \nabla f_i(\boldsymbol{x}_i^t) \right\|^2 \right)
$$

$$
\le \quad \frac{1}{2T} \left( 192k^2 \gamma^4 L^4 \sum_{t=k}^{T-1} \sum_{\tau'=t''}^{t'-1} \sum_{s=\tau'}^{t-1} \left\| \frac{1}{N} \sum_{i=1}^{N} \nabla f_i(\boldsymbol{x}_i^s) \right\|^2 - \sum_{t=0}^{T-1} \mathbb{E} \left\| \frac{1}{N} \sum_{i=1}^{N} \nabla f_i(\boldsymbol{x}_i^t) \right\|^2 \right)
$$

$$
\le \quad \frac{1}{2T} \left( 384k^4 \gamma^4 L^4 \sum_{t=0}^{T-1} \left\| \frac{1}{N} \sum_{i=1}^{N} \nabla f_i(\boldsymbol{x}_i^t) \right\|^2 - \sum_{t=0}^{T-1} \mathbb{E} \left\| \frac{1}{N} \sum_{i=1}^{N} \nabla f_i(\boldsymbol{x}_i^t) \right\|^2 \right)
$$

$$
\le \quad \frac{384k^4 \gamma^4 L^4 - 1}{2T} \sum_{t=0}^{T-1} \left\| \frac{1}{N} \sum_{i=1}^{N} \nabla f_i(\boldsymbol{x}_i^t) \right\|^2. \tag{49}
$$

Since $72k^2 \gamma^2 L^2 \le 1$, then we have $384k^4 \gamma^4 L^4 \le 1$, and thus $T_8 \le 0$. Rearranging (48) and dividing both sides by $\left( 1 - 24k^2 \gamma^2 L^2 \right)$, we get

$$
\frac{1}{T} \sum_{t=0}^{T-1} \mathbb{E} \| \nabla f(\hat{\boldsymbol{x}}^t) \|^2 \le \quad \frac{2(f(\hat{\boldsymbol{x}}^0) - f^*)}{T\gamma(1 - 24k^2\gamma^2 L^2)} + \frac{\gamma L \sigma^2}{N(1 - 24k^2\gamma^2 L^2)} + \frac{36k\gamma^2 \sigma^2 L^2}{1 - 24k^2\gamma^2 L^2}
$$

$$
+ \frac{8\gamma^2 L^2 C}{T(1 - 24k^2\gamma^2 L^2)} + \frac{96k^3 \gamma^4 \sigma^2 L^4}{N(1 - 24k^2\gamma^2 L^2)}
$$

$$
\le \quad \frac{3(f(\hat{\boldsymbol{x}}^0) - f^*)}{T\gamma} + \frac{3\gamma L \sigma^2}{2N} + 54k\gamma^2 \sigma^2 L^2 + \frac{12\gamma^2 L^2 C}{T} + \frac{2k\gamma^2 \sigma^2 L^2}{N}
$$

$$
\le \quad \frac{3(f(\hat{\boldsymbol{x}}^0) - f^*)}{T\gamma} + \frac{3\gamma L \sigma^2}{2N} + 56k\gamma^2 \sigma^2 L^2 + \frac{12\gamma^2 L^2 C}{T}, \tag{50}
$$

where the inequalities hold because $k^2 \gamma^2 L^2 \le \frac{1}{72}$ and $\frac{1}{1-24k^2\gamma^2 L^2} \le \frac{3}{2}$.

## D    PROOF OF COROLLARY 5.2

In this section, we give the proof of Corollary 5.2.

**Corollary 5.2** *Under Assumption* 1, *when the learning rate is set as* $\gamma = \frac{\sqrt{N}}{\sigma\sqrt{T}}$ *and the total number satisfies* $T \geq \frac{64N^3L^2k^2}{\sigma^2}$ *, we have the following convergence result for Algorithm 1:*

$$\frac{1}{T}\sum_{t=0}^{T-1}\mathbb{E}\left\|\nabla f(\hat{\boldsymbol{x}}^t)\right\| \leq \frac{3\sigma(f(\hat{\boldsymbol{x}}^0)-f^*+3L)}{\sqrt{NT}} + \frac{12NC}{\sigma^2T^2}. \tag{51}$$

*Proof.* Since $\gamma = \frac{\sqrt{N}}{\sigma\sqrt{T}}$, $T \geq \frac{72N^3L^2k^2}{\sigma^2} \geq \frac{72Nk^2L^2}{\sigma^2}$, we have $72\gamma^2k^2L^2 \leq 1$ and $\gamma \leq \frac{1}{2L}$ . Then we can have the result in (34) and get

$$\frac{1}{T}\sum_{t=0}^{T-1}\mathbb{E}\|\nabla f(\hat{\boldsymbol{x}}^t)\|^2 \leq \frac{3(f(\hat{\boldsymbol{x}}^0)-f^*)}{T\gamma} + \frac{3\gamma L\sigma^2}{2N} + 56k\gamma^2\sigma^2L^2 + \frac{12\gamma^2L^2C}{T}. \tag{52}$$

Combing $\gamma = \frac{\sqrt{N}}{\sigma\sqrt{T}}$, $k^2\gamma^2L^2 \leq \frac{1}{72}$ and $T \geq \frac{72N^3L^2k^2}{\sigma^2} \geq \frac{64N^3L^2k^2}{\sigma^2}$, we have

$$56k\gamma^2\sigma^2L^2 \quad \leq \quad 56k\frac{N}{\sigma^2T}\sigma^2L^2 = \frac{56kNL^2}{\sqrt{T}}\frac{1}{\sqrt{T}} \leq \frac{7\sigma L}{\sqrt{NT}}, \tag{53}$$

$$\frac{3\gamma L\sigma^2}{2N} \quad = \quad \frac{3\sigma L}{2\sqrt{NT}}, \tag{54}$$

$$\frac{3(f(\hat{\boldsymbol{x}}_0)-f^*)}{T\gamma} \quad = \quad \frac{3\sigma(f(\hat{\boldsymbol{x}}_0)-f^*)}{\sqrt{NT}}, \tag{55}$$

$$\frac{12\gamma^2L^2C}{T} \quad = \quad \frac{12NC}{\sigma^2T^2}. \tag{56}$$

We can get the final result

$$\frac{1}{T}\sum_{t=0}^{T-1}\mathbb{E}\left\|\nabla f(\hat{\boldsymbol{x}}^t)\right\| \leq \frac{3\sigma(f(\hat{\boldsymbol{x}}^0)-f^*+3L)}{\sqrt{NT}} + \frac{12NC}{\sigma^2T^2}, \tag{57}$$

which completes the proof.

# E MORE EXPERIMENTS

In this section, we evaluate the effectiveness of our algorithm on different variance among workers. Specifically, we consider the following finite-sum optimization

$$\min_{\boldsymbol{x}\in\mathbb{R}} f(\boldsymbol{x}) := \frac{1}{2}(f_1(\boldsymbol{x}) + f_2(\boldsymbol{x})) = 3\boldsymbol{x}^2 + 6b^2, \tag{58}$$

where $f_1(\boldsymbol{x}) := (x + 2b)^2$ and $f_2(\boldsymbol{x}) := 2(x - b)^2$ respectively denote the local loss function of the first and the second worker.

We can set a large variance among workers by adjusting b. Therefore, we can compare the convergence rate of algorithms in different variance, where the variance among workers is large with a large $b$. *VRL-SGD-W* denotes *VRL-SGD* with a warm-up, where the first communication period is set to 1. Figure 3 shows the gap with regard to iteration on different $k$ and $b$. We can see that *Local SGD* converges slowly compared with *VRL-SGD-W* and *VRL-SGD* when the communication period $k$ is relatively large. And *VRL-SGD* without warm-up is related to $b$ while *VRL-SGD-W* is not sensitive to $b$. Figure 4 shows that the variance of $v_i^t$ in *VRL-SGD* and *VRL-SGD-W* converges to 0, while the variance of $\nabla f_i(\boldsymbol{x})$ in *Local SGD* is a constant related to $b$. The experimental results verify our conclusion that VRL-SGD has a better convergence rate compared with *Local SGD* in *the non-identical case*, and *VRL-SGD* with a warm-up is more robustness to the variance among workers.

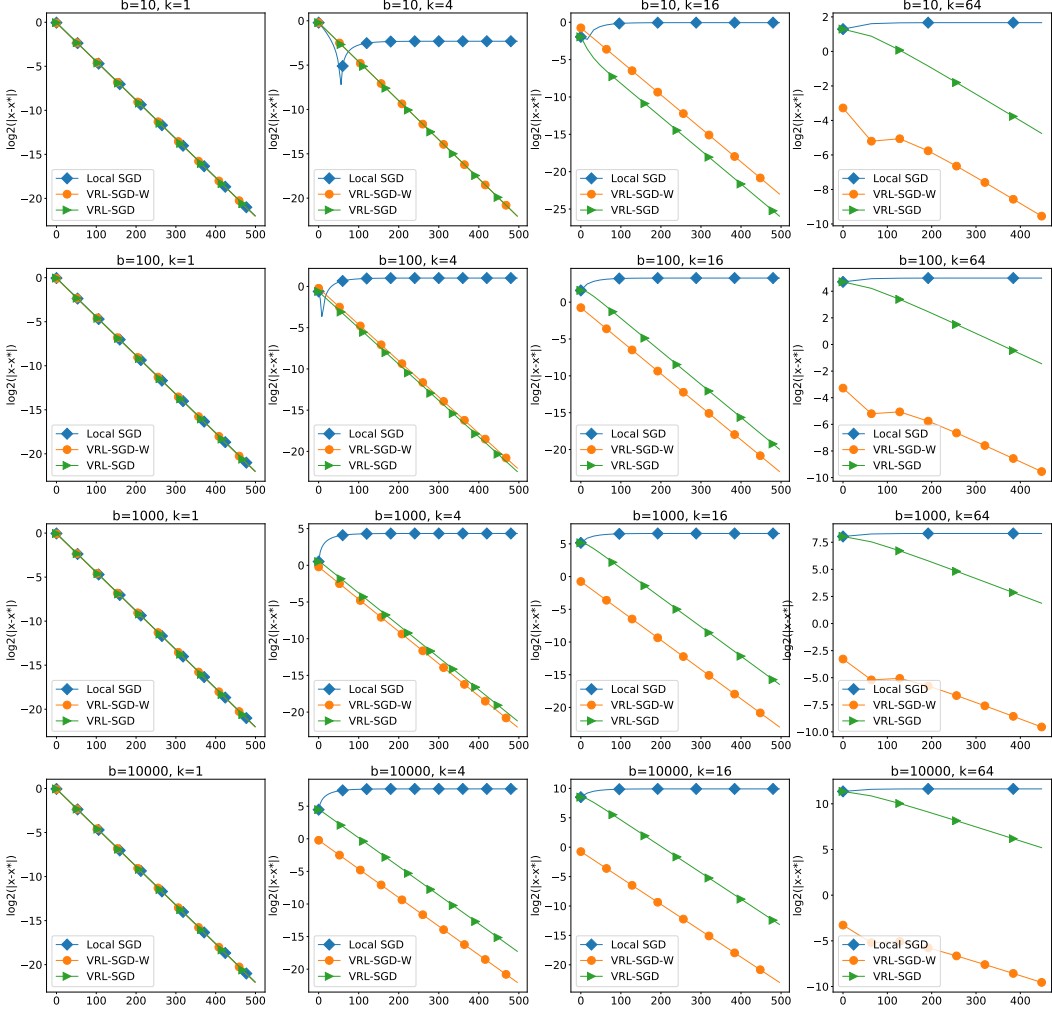

Figure 3: Logarithm of distance to the global minimum for different b and communication period k.

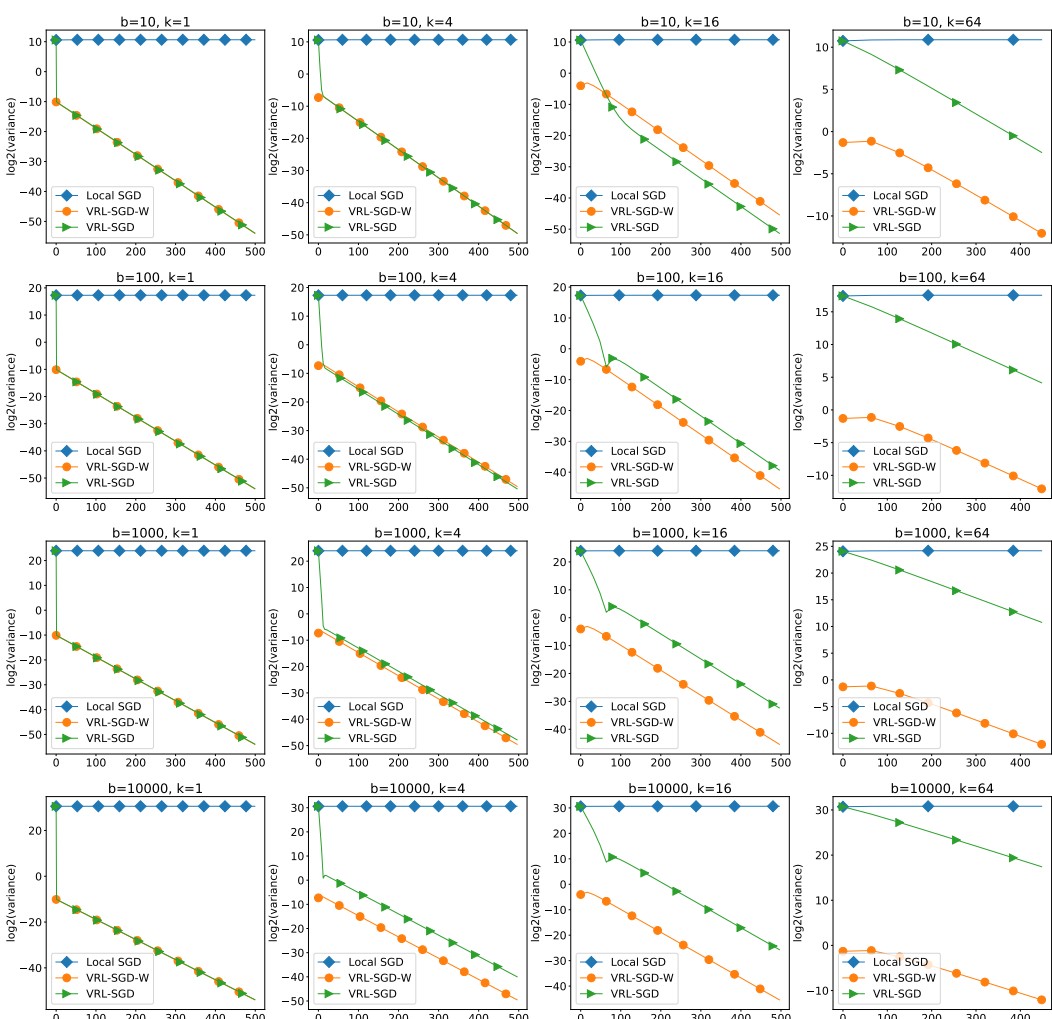

Figure 4: Logarithm of variance among workers for different b and communication period k.

## F    THE ANALYSIS OF PARAMETER K

In this section, we evaluate all algorithms with different communication period $k$.

As shown in Figure 5, VRL-SGD converges as fast as S-SGD, while Local SGD, EASGD converge slowly even if we set the period $k$ to half of it in Figure 1. The results show that $k$ in Local SGD should be smaller, such as $k = 2$ or $k = 5$ in transfer learning, which is in line with $\frac{T^{\frac{1}{4}}}{N^{\frac{3}{4}}} = \frac{117,187^{\frac{1}{4}}}{8^{\frac{3}{4}}} \approx 3.9$. However, we can set $k$ to $\frac{T^{\frac{1}{2}}}{N^{\frac{3}{2}}} = \frac{117,187^{\frac{1}{2}}}{8^{\frac{3}{2}}} \approx 15$ in VRL-SGD. Figure 6 compares the convergence of different algorithms with a larger $k$. We observe that the convergence of VRL-SGD will be affected with much large $k$, but VRL-SGD is still faster than Local SGD and EASGD, which is consistent with our theoretical analysis.

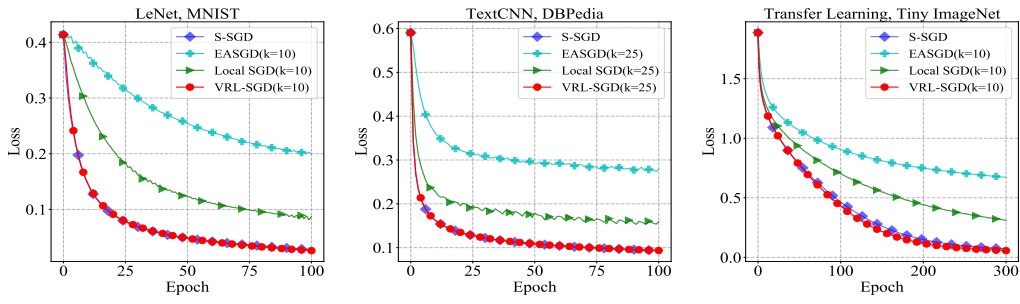

Figure 5: Epoch loss for the *non-identical case*. We set $k = 10$ for LeNet, $k = 25$ for TextCNN and $k = 10$ for Transfer Learning.

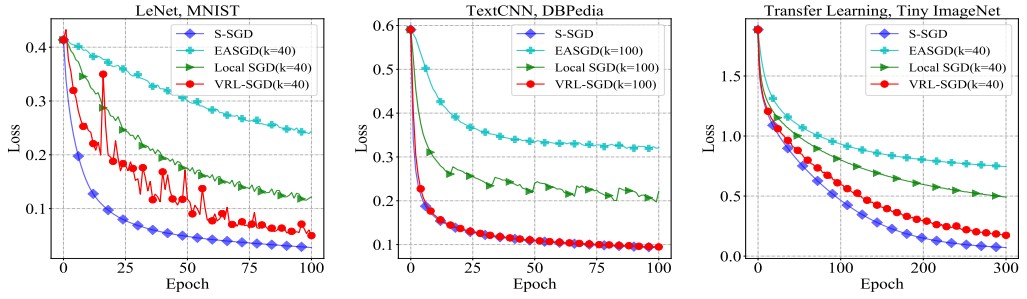

Figure 6: Epoch loss for the *non-identical case*. We set $k = 40$ for LeNet, $k = 100$ for TextCNN and $k = 40$ for Transfer Learning.

