# OpenReview forum: "Variance Reduced Local SGD with Lower Communication Complexity"
_ICLR.cc/2020/Conference — Reject_

### Official Review · AnonReviewer2 · 2019-10-18
**Official Blind Review #2**

**Rating:** 3

**Review:**

The paper proposes the variance reduction to the local SGD algorithm and shows the proposed VRL-SGD can achieve better convergence than local SGD when the data are not identical over workers. The idea is interesting and the paper is easy to follow. However, the study does not go into depth. I do not support the acceptance for current situation.

1. The paper does not show the convergence rate when the function is convex/strongly convex, which make it hard to compare with previous works e.g. EXTRA [Shi et al. 2015].
2. There are many alternatives to achieve communication efficiency. The paper does not argue why to choose variance reduced Local SGD. A natural idea to reduce the variance is to distribute the inner loop of SVRG to different workers as proposed in [Konecny et al. 2016]. Moreover, the analysis is given in [Lee et al. 2015] and [Cen et al. 2019]. Especially, [Cen et al. 2019] suggests using a regularization term to handle the data load is not balanced over the workers.
3. The paper states that S-SGD cannot achieve linear iteration speedup due to communication bottleneck. Can we avoid the communication bottleneck by increasing the batch size? There are vast literatures on distributed training of deep neural network by using large batch size.
4. The experiment comparison is not complete given there are many related work in this area.

Question: How to determine the number of local SGD steps for each communication round? In SVRG, the number of iterations in the inner loop is related to the condition number (strongly convex case). Does the number of local SGD steps have a similar correspondence

[Jason Lee, et al.]  2015. Distributed Stochastic Variance Reduced Gradient Methods and A Lower Bound for Communication Complexity.
[Shicong Cen, et al.] 2019 Convergence of Distributed Stochastic Variance Reduced Methods without Sampling Extra Data


**Experience Assessment:**

I have published one or two papers in this area.

**Review Assessment: Checking Correctness Of Derivations And Theory:**

I assessed the sensibility of the derivations and theory.

**Review Assessment: Checking Correctness Of Experiments:**

I carefully checked the experiments.

**Review Assessment: Thoroughness In Paper Reading:**

I read the paper at least twice and used my best judgement in assessing the paper.

---

> ### Author Response · Authors · 2019-11-11
> **Response to Review #2**
>
> Dear Reviewer 2,
> Thanks for reviewing our paper. Our replies to your reviews are as follows:
> 1.	Our work mainly focuses on non-convex stochastic optimization. The theoretical analysis in convex/strong convex is also meaningful, and we will consider it in future work. It is unfair and hard to compare the convergence of VRL-SGD with EXTRA, because EXTRA is a deterministic algorithm, while VRL-SGD is a stochastic one. In the latest version (Remark 5.4), we compare the convergence of VRL-SGD with D^2, which is a stochastic variant of EXTRA. VRL-SGD has a tighter convergence rate and a lower communication complexity compared with D^2.
>
> 2.	a) Among the communication efficiency algorithms, Local SGD is simple and has great theoretical properties. We mainly focus on improving the performance of Local SGD in the non-iid case.
>        b) The variance reduction techniques in SVRG and its variants are used to reduce the variance caused by stochastic sampling, while they are time-consuming or memory consuming when applied in a distributed setting, especially training deep learning models. This is because that they need to calculate full gradients or store historical gradients. We use a lightweight variant of the variance reduction technique in SVRG and achieve the goal of eliminating the variance caused by non-iid without more calculation or memory.
>
> 3.	It had been reported that too large mini-batch sizes would cause a poor generalization in previous studies [1,2,3].
> [1] Nitish Shirish Keskar, Dheevatsa Mudigere, Jorge Nocedal, Mikhail Smelyanskiy, Ping Tak Peter Tang: On Large-Batch Training for Deep Learning: Generalization Gap and Sharp Minima. ICLR 2017
> [2] Siyuan Ma, Raef Bassily, Mikhail Belkin: The Power of Interpolation: Understanding the Effectiveness of SGD in Modern Over-parametrized Learning. ICML 2018
> [3] Dong Yin, Ashwin Pananjady, Maximilian Lam, Dimitris S. Papailiopoulos, Kannan Ramchandran, Peter L. Bartlett: Gradient Diversity: a Key Ingredient for Scalable Distributed Learning. AISTATS 2018
>
> 4.	Thank you for your suggestion to improve our experiments, we have 	added more baselines, such as EASGD mentioned by Reviewer 3. There are many algorithms based on Local SGD, but most of them focus on reducing communication costs, not addressing the issue in the non-iid case.
>
> The answer to your question: as claimed in Remark 5.6, it suffices to choose any k\le O(T^{\frac{1}{2}}/N^{\frac{3}{2}}).

---

### Official Review · AnonReviewer1 · 2019-10-22
**Official Blind Review #1**

**Rating:** 1

**Review:**

In the paper, the authors propose a variance reduced local SGD and prove its convergence rate. In the experiments, they show that the proposed method converges faster than local SGD.

The following are my concerns:
1) I am concerned about the convergence result, it shows that the convergence of the proposed method has nothing to do with the extent of non-iid. However, it is not correct intuitively. It is easy to imagine that non-iid data will converge slower than iid data.

2) In Corollary 5.2, the convergence result is not related to k. It is false to me.

3) It is not clear in algorithm 1 how the \delta^{t''} is updated.

4) The assumption in equation (11)  "When all local model x^t, x\tau and the average model \hat x converge to the local minimum x∗" is not correct when data is non-iid distributed. Suppose x^t and \hat x is x^*,  and \Delta^{t''} = 0.  Because data is non-iid, the solution of the local problem is not equal to the global problem, therefore, x^t will go away from x^*.

5) In the experiment, the setting of k should affect the experiment. However, authors don't analyze this parameter.





**Experience Assessment:**

I have published in this field for several years.

**Review Assessment: Checking Correctness Of Derivations And Theory:**

I assessed the sensibility of the derivations and theory.

**Review Assessment: Checking Correctness Of Experiments:**

I assessed the sensibility of the experiments.

**Review Assessment: Thoroughness In Paper Reading:**

I read the paper at least twice and used my best judgement in assessing the paper.

---

> ### Author Response · Authors · 2019-11-11
> **Response to Review #1**
>
> Dear Reviewer 1,
> Thanks for reviewing our paper.
>
> Here are our responses that would clarify your concerns.
> 1)	A)  Compared with Local SGD, VRL-SGD uses an extra variance reduction technique which can eliminate the gradient variance among workers. Thus, the impact caused by non-iid is removed in our result, which is our main contribution. A similar result can be found in existing study [1], where non-iid data converge with the same speed as iid data.
> B)  Sorry for the confusion. There was an error in the submitted version since we ignored the first period, where VRL-SGD was consistent with Local SGD without variance reduction. We have fixed this problem in the latest submission.
> C)  We also have added some additional experiments in Appendix E to support our conclusion. As shown in Figure 4 in the latest submission, the gradient variance among workers in VRL-SGD is significantly smaller than it in Local SGD. The variance in VRL-SGD asymptotically tends to zero while it is always a constant in Local SGD.
> 2)	As shown in Corollary 5.2, VRL-SGD has the convergence rate O(1/sqrt{NT}) conditioned on T \ge (72N^3L^2K^2)/(sigma^2). This means that, when NT is fixed, to achieve this convergence rate, k needs to satisfy k \le O(T^{1/2}/N^{3/2}). Therefore, the convergence result is still related to k.
> 3)	According to 3.2 NOTATIONS, t`` = t`-k. Therefore \Delta^{t''}_i and \Delta^{t'}_i represent the same variable \Delta_i of the i-th worker at different time. As shown in Eq (4), the update form of \Delta_i is: \Delta_i += 1/(k\gamma) (\hat{x}^t – x_i^t).
> 4)	We do not agree with you. Note that \Delta_{i}^t is not zero in the above supposition. When x^\tau and \hat x are x^*, from the new representation shown of \Delta_i in Eq (9) we can see that \Delta_i^t = \nabla f_i(x^*) – 1/N \sum_{j=1}^N \nablaf_j(x^*) = \nabla f_i(x^*) - \nabla f(x^*) = \nabla f_i(x^*) \neq 0. Besides, one can see from Figure 3 in the latest version, the model in VRL-SGD converges to the optimal solution even the data is non-iid.
> 5)	A)	Thank you for your suggestion to improve our experiments, we have add more experimental results to analyze the influence of parameter k in Appendix F.
> B)	In Remark 5.4, we have analyzed that the parameter k needs to satisfy k\le O(T^{\frac{1}{2}}/N^{\frac{3}{2}}) to guarantee the linear speedup of VRL-SGD. So it suffices to choose any k\le O(T^{\frac{1}{2}}/N^{\frac{3}{2}}).
>
>  We are sincerely expected that you can take them into account when making the final recommendation. Great Thanks!
>
> [1] Hanlin Tang, Xiangru Lian, Ming Yan, Ce Zhang, and Ji Liu. D^2 : Decentralized training over decentralized data. In ICML, pp. 4855–4863, 2018.

---

### Official Review · AnonReviewer3 · 2019-10-24
**Official Blind Review #3**

**Rating:** 6

**Review:**

This paper tackles the problem of data-parallel (synchronous) distributed SGD, to optimize the (finite) sum of N non-convex, possibly different (in the so-called non-identical case), loss functions. This paper focuses on improving the communication efficiency compared to several existing methods tackling this problem.

To that end, the authors contribute:
· A novel algorithm and its asymptotic communication complexity.
· The proof that the common metric of the sum over the training steps of the expected squared norm of the gradient at the average of the N parameters is bounded above.
· Experimental results (training loss function of epoch number) comparing this algorithm with 2 existing ones, solving 3 problems under reasonable settings.

  The training time per epoch of VRL-SGD is claimed to be identical to the one of Local SGD, as the algorithm only have minor differences.

- strengths of the paper:

· The main paper is very easy to follow.
· Good effort to give intuitions on why VLR-SGD can improve the convergence rate of existing algorithms.
  Such an effort is to be highlighted.
· No obvious mistake in the main.   I have not thoroughly checked the full proof though.

-  weaknesses of the paper:

· The algorithm, while having differences, is quite reminiscent of Elastic Averaging SGD (EASGD) [1].
  Indeed in both algorithms the model update at the workers consists in both descending the local gradient plus descending toward some "moving-average"obtained through averaging all the local models.
  In EASGD, this "moving-average" is common to every worker and the master, which updates it every k steps.
  In this paper, each worker has its own "moving-average", which update computations are different than in EASGD as the use the instant average of the workers' models instead of the previous "moving-average".

[1]Sixin Zhang, Anna Choromanska, Yann LeCun, Deep learning with Elastic Averaging SGD,  NeurIPS, 2015

- Questions I would like the authors to respond to during the rebuttal:

· Could Elastic Averaging SGD (in particular their fastest variant EAMSGD) be applied as-is to solve the non-identical, non-convex optimization problem at hand?
  Despite the authors of EASGD not studying their algorithm in the non-identical case, following what is done in the intuition part of VRL-SGD (in particular Equation (8)), it seems that the update rule of the "moving-average" in EASGD is then equivalent to having a momentumSGD with dampening (instead of the "generalized SGD form" obtained with the approach of VRL-SGD).  Hence my question.

I suggest acceptance. However I'm willing to change my opinion after reading other more qualified reviewers in the sub-area of variance-reduction techniques.

note: If EASGD was to be sound in the non-identical case as well, my decision would not change much.

**Experience Assessment:**

I have published one or two papers in this area.

**Review Assessment: Checking Correctness Of Derivations And Theory:**

I assessed the sensibility of the derivations and theory.

**Review Assessment: Checking Correctness Of Experiments:**

I assessed the sensibility of the experiments.

**Review Assessment: Thoroughness In Paper Reading:**

I read the paper at least twice and used my best judgement in assessing the paper.

---

> ### Author Response · Authors · 2019-11-11
> **Response to Review #3**
>
> Dear Reviewer 3,
>
> Thank you for your suggestion to improve our experiments.
>
> We have added the comparison between EASGD and VRL-SGD in the latest version. And the experimental results show that EASGD is not as good as VRL-SGD in non-iid case, which is reasonable in our opinions with the following reasons.
> 1)	The local model and the global model in EASGD will never be consistent, thus EASGD is hard to converge in non-iid case.
> 2)	Elastic Force does not alleviate the bias of local gradient. The local gradient in EASGD is biased as Local SGD in the non-iid case. And the “Elastic Force” operation is only executed once in k updates, which is a special form of model averaging.
> 3)	To our best knowledge, there is no theoretical analysis for the convergence of asynchronous EASGD in the general non-convex setting.

---

### Decision · Program_Chairs · 2019-12-19

**Decision:**

Reject

**Comment:**

The paper presents a novel variance reduction algorithm for SGD. The presentation is clear. But the theory is not good enough. The reivewers worry about the converge results and the technical part is not sound.